# LLM-Pruner: On the Structural Pruning of Large Language Models

**Xinyin Ma**    **Gongfan Fang**    **Xinchao Wang**[*]
National University of Singapore
`maxinyin@u.nus.edu, gongfan@u.nus.edu, xinchao@nus.edu.sg`

## Abstract

Large language models (LLMs) have shown remarkable capabilities in language understanding and generation. However, such impressive capability typically comes with a substantial model size, which presents significant challenges in both the deployment, inference, and training stages. With LLM being a general-purpose task solver, we explore its compression in a task-agnostic manner, which aims to preserve the multi-task solving and language generation ability of the original LLM. One challenge to achieving this is the enormous size of the training corpus of LLM, which makes both data transfer and model post-training over-burdensome. Thus, we tackle the compression of LLMs within the bound of two constraints: being task-agnostic and minimizing the reliance on the original training dataset. Our method, named LLM-Pruner, adopts structural pruning that selectively removes non-critical coupled structures based on gradient information, maximally preserving the majority of the LLM's functionality. To this end, the performance of pruned models can be efficiently recovered through tuning techniques, LoRA, in merely *3 hours*, requiring only *50K* data. We validate the LLM-Pruner on three LLMs, including LLaMA, Vicuna, and ChatGLM, and demonstrate that the compressed models still exhibit satisfactory capabilities in zero-shot classification and generation. The code is available at: `https://github.com/horseee/LLM-Pruner`

## 1   Introduction

Recently, Large Language Models (LLMs) [39, 51, 50, 44, 64, 4, 74] have demonstrated remarkable proficiency in language understanding and generation. With the increase in model size, they are better equipped to handle complex tasks  [3, 5, 58, 60] and even exhibit emergent abilities [57]. However, notwithstanding their impressive performance, LLMs pose challenges in deployment and inference. Their extensive scale engenders substantial computational demands, and the multitude of parameters involved can induce long latencies and other related issues. Several techniques are proposed to solve these problems, like model pruning [56, 61, 72, 21], knowledge distillation [46, 41, 47],quantization [1, 13] within the context of pre-trained language model (PLM).

While previous methods have effectively maintained model performance amidst parameter reduction, they primarily target compression within specialized domains or for designated tasks in the context of task-specific compression. For instance, a PLM is fine-tuned on a particular dataset, such as one of the classification tasks in the GLUE benchmark [53], after which these models are distilled into a smaller classification model [46, 18]. Although this paradigm could potentially be employed for LLM compression, it compromises the LLM's capacity as a versatile task solver, rendering it suited to a single task exclusively.

---

[*]Corresponding author

37th Conference on Neural Information Processing Systems (NeurIPS 2023).

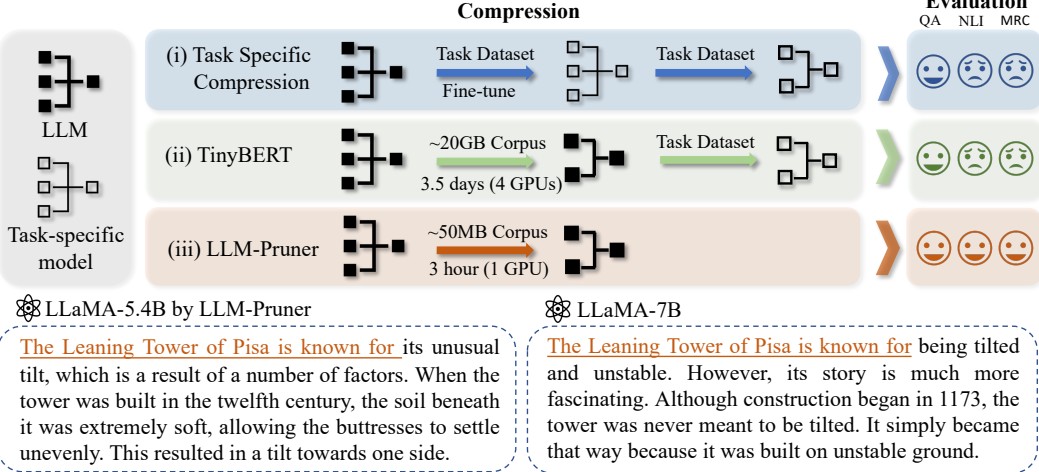

Figure 1: Illustration of LLM-Pruner. (i) Task-specific compression: the model is fine-tuned then compressed on a specific task. (ii) TinyBERT: First distill the model on unlabeled corpus and then fine-tune it on the specific task. (iii) LLM-Pruner: Task-agnostic compression within 3 hours.

Thus, we strive to compress the LLM in a new setting: to reduce the LLM's size while preserving its diverse capabilities as general-purpose task solvers, as depicted in Figure 1. This introduces the task-agnostic compression of LLMs, which presents two key challenges:

- **The size of the training corpus of the LLM is enormous.** Previous compression methods heavily depend on the training corpus. The LLM has escalated the corpus scale to 1 trillion tokens or more [17, 51]. The extensive storage needs and protracted transmission times make the dataset difficult to acquire. Furthermore, if the dataset is proprietary, acquisition of the training corpus verges on impossibility, a situation encountered in [74, 39].
- **The unacceptably long duration for the post-training of the pruned LLM.** Existing methods require a substantial amount of time for post-training the smaller model [55, 28]. For instance, the general distillation in TinyBERT takes around 14 GPU days [20]. Even post-training a task-specific compressed model of BERT demands around 33 hours [61, 22]. As the size of both the model and corpus for LLMs increases rapidly, this step will invariably consume an even more extensive time.

To tackle the aforementioned challenges associated with the task-agnostic compression of LLMs, we introduce a novel approach called LLM-Pruner. Since our goal is to compress LLMs with reduced data dependency and expedited post-training, how to prune model with the minimal disruption to the origin is crucial. To accomplish this, we propose a dependency detection algorithm that identifies all the dependent structures within the model. Once the coupled structure is identified, we employ an efficient importance estimation strategy to select the optimal group for pruning under the task-agnostic setting, where the first-order information and an approximated hessian information is taken into account. Finally, a rapid recovery stage is executed to post-train the pruned model with limited data.

**Contribution.** In this paper, we propose a novel framework, LLM-Pruner, for the task-agnostic compression of the large language model. To the best of our knowledge, LLM-Pruner is the first framework designed for structured pruning of LLMs. We conclude the advantages of the LLM-Pruner as (i) Task-agnostic compression, where the compressed language model retains its ability to serve as a multi-task solver. (ii) Reduced demand for the original training corpus, where only 50k publicly available samples are needed for compression, significantly reducing the budget for acquiring the training data (iii) Quick compression, where the compression process ends up in three hours. (iv) An automatic structural pruning framework, where all the dependent structures are grouped without the need for any manual design. To evaluate the effectiveness of LLM-Pruner, we conduct extensive experiments on three large language models: LLaMA-7B, Vicuna-7B, and ChatGLM-6B. The compressed models are evaluated using nine datasets to assess both the generation quality and the zero-shot classification performance of the pruned models. The experimental results demonstrate that even with the removal of 20% of the parameters, the pruned model maintains 94.97% of the performance of the original model.

# 2   Related Work

**Compression of Language Model.**   Language models [9, 31, 25] have gained much attention and increase the need to reduce the size of parameters and reduce the latency [23, 48]. To compress the language model, previous works can be divided into several categories: network pruning [21, 63, 32, 15], knowledge distillation [46, 47, 40], quantization [68, 1, 71] and other techniques, like early exit [62] or dynamic token reduction [69]. We focus on the pruning of the language models, especially structural pruning [26]. Structural pruning removes the entire filter from the neural network, which is more hardware friendly. There are several ways to remove the structure, such as l1-dependent pruning [16, 72], first-order importance estimation [18], hessian-based estimation [21, 54] or the optimal brain surgeon [24, 21]. As for the pruning unit in structural pruning, some works adopt the entire layer [10] as the minimal unit, and others take the multi-head attention [52] or the feed-forward layers [18, 36] as the basic structure to prune. CoFi [61] studies the pruning unit in different granularity.

**Efficient and Low Resource Compression.**   With the growing size of neural network models, there is an increasing demand for efficient and low-resource compression [67, 66, 30, 29, 65]. As for the efficient compression, [22] accelerate the post-training by defining the reconstruction error as a linear least squares problem. [13, 12] propose the layer-wise optimal brain surgeon. As for the constraint of availability of the training corpus, data-free pruning [45, 70] come up with several strategies to prune the model by measuring neurons' similarity. Besides, [34, 33, 42] proposes methods that distill the model without reliance on the training corpus of the model. However, those methods are too time-consuming, involving synthesizing samples by backpropagating the pre-trained models.

# 3   Methods

In this section, we provide a detailed explanation of LLM-Pruner. Following the conventional model compression pipeline[22], LLM-Pruner consists of three steps: **(1) Discovery Stage** (Section 3.1). This step focuses on identifying groups of interdependent structures within LLMs. **(2) Estimation Stage** (Section 3.2). Once the coupled structures are grouped, the second step entails estimating the contribution of each group to the overall performance of the model and deciding which group to be pruned. **(3) Recover Stage** (Section 3.3). This step involves fast post-training that alleviates potential performance degradation caused by the removal of structures.

## 3.1   Discover All Coupled Structure in LLMs

In light of the limited availability of data for post-training, it becomes imperative to prioritize the removal of structures with minimal damage when compressing the model. This underscores the dependency-based structural pruning, which ensures coupled structures are pruned in unison. We provide an experiment in Section 4.3 to show the importance of dependency-based structural pruning when compressing the large language model.

**Structure Dependency in LLMs.**   Similar to [11], the pruning begins by building the dependency for LLMs. Assume $N_i$ and $N_j$ are two neurons in the model, $\text{In}(N_i)$ and $\text{Out}(N_i)$ represents all the neurons that point towards or point from $N_i$. The dependency between structures can be defined as:

$$N_j \in \text{Out}(N_i) \wedge \text{Deg}^-(N_j) = 1 \Rightarrow N_j \text{ is dependent on } N_i \tag{1}$$

where $\text{Deg}^-(N_j)$ represents the in-degree of neuron $N_j$. Noting that this dependency is directional, we can therefore correspondingly obtain another dependency:

$$N_i \in \text{In}(N_j) \wedge \text{Deg}^+(N_i) = 1 \Rightarrow N_i \text{ is dependent on } N_j \tag{2}$$

where $\text{Deg}^+(N_i)$ represents the out-degree of neuron $N_i$. The principle of dependency here is, if a current neuron (e.g., $N_i$) depends solely on another neuron (e.g., $N_j$), and the neuron $N_j$ is subjected to pruning, it follows that the neuron $N_i$ must also undergo pruning. We provide a detailed case in Appendix A.

**Trigger the Dependency Graph.**   By having the definition of dependency, the coupled structures in the LLM can be analyzed automatically. Considering any neuron within the LLM as the initial

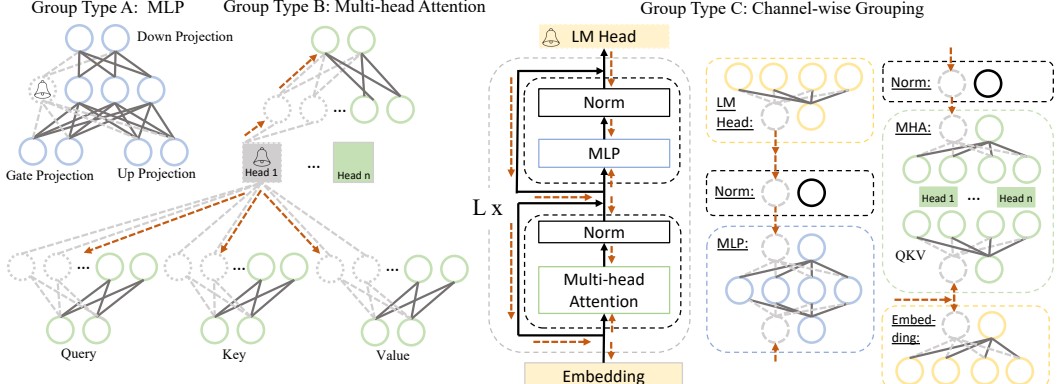

Figure 2: Illustration of the coupled structures in LLaMA. We simplify the neurons in each layer to make the dependent group clear. The trigger neuron, marked as a circle with a bell, cause weights with dependency pruned (dashed lines), which may propagate (red dashed lines) to coupled neurons (dashed circles). A group can be triggered by a variety of trigger neurons. Taking Group Type B as an example, the trigger for this group involves (i) the attention head, (ii) the output neuron in Query, Key or Value, and (iii) the input neuron in the final output projection.

trigger, it possesses the capability to activate neurons that depend on it. Subsequently, these newly triggered neurons can serve as the subsequent triggers to identify the dependency and activate their respective dependent neurons. This iterative process continues until no new neurons are detected. Those neurons then form a group for further pruning. Taking LLaMA as an example, by searching over all the neurons as the initial trigger, we can locate all the coupled structures, as shown in Figure2.

Given the diversity in the structure of different LLMs, manual analysis and removal of coupled structures in each LLM could be extremely time-consuming. However, by employing LLM-Pruner, all coupled structures can be automatically identified and extracted.

### 3.2 Grouped Importance Estimation of Coupled Structure

Till now, all coupled structures within the model are grouped. Weights within the same group should be pruned simultaneously, as partial pruning not only increases parameter size but also introduces misaligned intermediate representations. Therefore, we estimate the importance of the group as a whole, as opposed to evaluating the importance of modules. Given the limited access to the training dataset, we explore the use of public datasets or manually created samples as alternative resources. Although the domains of these datasets may not perfectly align with the training set, they still provide valuable information for assessing the importance.

**Vector-wise Importance.** Suppose that given a dataset $\mathcal{D} = \{x_i, y_i\}_{i=1}^{N}$, where N is the number of samples. In our experiments, we set N equal to 10 and we use some public datasets as the source of $\mathcal{D}$. A group (as previously defined as a set of coupled structures) can be defined as $\mathcal{G} = \{W_i\}_{i=1}^{M}$, where M is the number of coupled structures in one group and $W_i$ is the weight for each structure. While pruning, our goal is to remove the group that has the least impact on the model's prediction, which can be indicated by the deviation in the loss. Specially, to estimate the importance of $W_i$, the change in loss can be formulated as [24]:

$$I_{W_i} = |\Delta \mathcal{L}(\mathcal{D})| = |\mathcal{L}_{W_i}(\mathcal{D}) - \mathcal{L}_{W_i=0}(\mathcal{D})| = |\underbrace{\frac{\partial \mathcal{L}^\top(\mathcal{D})}{\partial W_i}}_{\neq 0} W_i - \frac{1}{2} W_i^\top H W_i + \mathcal{O}\left(\|W_i\|^3\right)| \quad (3)$$

where $H$ is the hessian matrix. Here, $\mathcal{L}$ represents the next-token prediction loss. The first term is typically neglected in prior work [24, 54, 12], as the model has already converged on the training dataset, where $\partial \mathcal{L}^\top / \partial W_i \approx 0$. However, since $\mathcal{D}$ here is not extracted from the original training data, which means that $\partial \mathcal{L}^\top / \partial W_i \not\approx 0$. This presents a desirable property for determining the importance of $W_i$ by the gradient term under LLMs, since computation of the second term, the Hessian matrix, on the LLM is impractical with $\mathcal{O}\left(N^2\right)$ complexity.

**Element-wise Importance.** The above can be considered as an estimate for the weight $W_i$. We can derive another measure of importance at a finer granularity, where each parameter within $W_i$ is assessed for its significance:

$$I_{W_i^k} = |\Delta \mathcal{L}(\mathcal{D})| = |\mathcal{L}_{W_i^k}(\mathcal{D}) - \mathcal{L}_{W_i^k=0}(\mathcal{D})| = |\frac{\partial \mathcal{L}(\mathcal{D})}{\partial W_i^k} W_i^k - \frac{1}{2} W_i^k H_{kk} W_i^k + \mathcal{O}\left(\|W_i^k\|^3\right)| \quad (4)$$

Here, $k$ represents the k-th parameter in $W_i$. The diagonal of the hessian $H_{kk}$ can be approximated by the Fisher information matrix, and the importance can be defined as:

$$I_{W_i^k} = |\mathcal{L}_{W_i^k}(\mathcal{D}) - \mathcal{L}_{W_i^k=0}(\mathcal{D})| \approx |\frac{\partial \mathcal{L}(\mathcal{D})}{\partial W_i^k} W_i^k - \frac{1}{2} \sum_{j=1}^{N} \left( \frac{\partial \mathcal{L}(\mathcal{D}_j)}{\partial W_i^k} W_i^k \right)^2 + \mathcal{O}\left(\|W_i^k\|^3\right)| \quad (5)$$

**Group Importance.** By utilizing either $I_{W_i^k}$ or $I_{W_i}$, we estimate the importance at the granularity of either a parameter or a vector of weight. Remembering that our goal is to estimate the importance of $\mathcal{G}$, we aggregate the importance scores in four ways: (i) Summation: $I_{\mathcal{G}} = \sum_{i=1}^{M} I_{W_i}$ or $I_{\mathcal{G}} = \sum_{i=1}^{M} \sum_k I_{W_i^k}$, (ii) Production: $I_{\mathcal{G}} = \prod_{i=1}^{M} I_{W_i}$ or $I_{\mathcal{G}} = \prod_{i=1}^{M} \sum_k I_{W_i^k}$, (iii) Max: $I_{\mathcal{G}} = \max_{i=1}^{M} I_{W_i}$ or $I_{\mathcal{G}} = \max_{i=1}^{M} \sum_k I_{W_i^k}$; (iv) Last-Only: Since deleting the last executing structure in a dependency group is equivalent to erasing all the computed results within that group, we assign the importance of the last executing structure as the importance of the group: $I_{\mathcal{G}} = I_{W_l}$ or $I_{\mathcal{G}} = \sum_k I_{W_l^k}$, where $l$ is the last structure. After assessing the importance of each group, we rank the importance of each group and prune the groups with lower importance based on a predefined pruning ratio.

### 3.3 Fast Recovery with Low-rank Approximation

In order to expedite the model recovery process and improve its efficiency under limited data, it is crucial to minimize the number of parameters that need optimization during the recovery phase. To facilitate this, we employ the low-rank approximation, LoRA[19], to post-train the pruned model. Each learnable weight matrix in the model, denoted as $W$, encompassing both pruned and unpruned linear projection in the LLM, can be represented as $W$. The update value $\Delta W$ for $W$ can be decomposed as $\Delta W = PQ \in \mathbb{R}^{d^- \times d^+}$, where $P \in \mathbb{R}^{d^- \times d}$ and $Q \in \mathbb{R}^{d \times d^+}$. The forward computation can now be expressed as:

$$f(x) = (W + \Delta W)X + b = (WX + b) + (PQ)X \quad (6)$$

where $b$ is the bias in the dense layer. Only training $P$ and $Q$ reduces the overall training complexity, reducing the need for large-scale training data. Besides, the extra parameters $P$ and $Q$ can be reparameterized into $\Delta W$, which would not cause extra parameters in the final compressed model.

## 4 Experiments

### 4.1 Experimental Settings

**Foundation Large Language Model.** To showcase the effectiveness and versatility of LLM-Pruner, we test it over three open-source large language models with two kinds of structure: LLaMA-7B [51], Vicuna-7B [4] [2] and ChatGLM-6B [74].

**Evaluation and Datasets.** To assess the performance of the model in the task-agnostic setting, we follow LLaMA's evaluation to perform zero-shot task classification on common sense reasoning datasets: BoolQ [6], PIQA [2], HellaSwag [73], WinoGrande [43], ARC-easy [7], ARC-challenge [7] and OpenbookQA [38]. Follow [14], the model ranks the choices in the multiple choice tasks or generates the answer in the open-ended generation [3]. Additionally, we complement our evaluation with a zero-shot perplexity (PPL) analysis on WikiText2 [37] and PTB [35].

---

[2]https://huggingface.co/lmsys/vicuna-7b-delta-v0
[3]https://github.com/EleutherAI/lm-evaluation-harness

Table 1: Zero-shot performance of the compressed LLaMA-7B. The average is calculated among seven classification datasets. 'Underline' indicates the best pruning-only performance, while 'bold' represents the overall best performance with the same pruning ratio, considering both pruning and post-training. The 'Channel' strategy only prunes the dependent group of Type C, while all other methods employ the 'Block' strategy to prune dependent groups in both Type A and Type B. Since [51] did not provide its prompt, the evaluation of the result with ⋆ is performed under different prompts, which is lower than the official results.

| Pruning Ratio | Method | WikiText2↓ | PTB↓ | BoolQ | PIQA | HellaSwag | WinoGrande | ARC-e | ARC-c | OBQA | Average |
|---|---|---|---|---|---|---|---|---|---|---|---|
| Ratio = 0% | LLaMA-7B[51] | - | - | 76.5 | 79.8 | 76.1 | 70.1 | 72.8 | 47.6 | 57.2 | 68.59 |
| | LLaMA-7B⋆ | 12.62 | 22.14 | 73.18 | 78.35 | 72.99 | 67.01 | 67.45 | 41.38 | 42.40 | 63.25 |
| Ratio = 20% w/o tune | L2 | 582.41 | 1022.17 | 59.66 | 58.00 | 37.04 | 52.41 | 33.12 | 28.58 | 29.80 | 42.65 |
| | Random | 27.51 | 43.19 | 61.83 | 71.33 | 56.26 | 54.46 | 57.07 | 32.85 | 35.00 | 52.69 |
| | Channel | 74.63 | 153.75 | 62.75 | 62.73 | 41.40 | 51.07 | 41.38 | 27.90 | 30.40 | 45.38 |
| | Vector | 22.28 | 41.78 | 61.44 | 71.71 | 57.27 | 54.22 | 55.77 | 33.96 | 38.40 | 53.25 |
| | Element[2] | 19.77 | 36.66 | 59.39 | 75.57 | 65.34 | 61.33 | 59.18 | 37.12 | 39.80 | 56.82 |
| | Element[1] | 19.09 | 34.21 | 57.06 | 75.68 | 66.80 | 59.83 | 60.94 | 36.52 | 40.00 | 56.69 |
| Ratio = 20% w/ tune | Channel | 22.02 | 38.67 | 59.08 | 73.39 | 64.02 | 60.54 | 57.95 | 35.58 | 38.40 | 55.57 |
| | Vector | 18.84 | 33.05 | 65.75 | 74.70 | 64.52 | 59.35 | 60.65 | 36.26 | 39.40 | 57.23 |
| | Element[2] | 17.37 | 30.39 | 69.54 | 76.44 | 68.11 | 65.11 | 63.43 | 37.88 | 40.00 | 60.07 |
| | Element[1] | 17.58 | 30.11 | 64.62 | 77.20 | 68.80 | 63.14 | 64.31 | 36.77 | 39.80 | 59.23 |

Table 2: Zero-shot performance of the compressed LLaMA-13B. Here we adopt Element[1] as the importance estimation for 'Channel' and 'Block'.

| Pruning Ratio | Method | WikiText2↓ | PTB↓ | BoolQ | PIQA | HellaSwag | WinoGrande | ARC-e | ARC-c | OBQA | Average |
|---|---|---|---|---|---|---|---|---|---|---|---|
| Ratio = 0% | LLaMA-13B⋆ | 11.58 | 20.24 | 68.47 | 78.89 | 76.24 | 70.09 | 74.58 | 44.54 | 42.00 | 64.97 |
| Ratio = 20% w/o tune | L2 | 61.15 | 91.43 | 61.50 | 67.57 | 52.90 | 57.54 | 50.13 | 31.14 | 36.80 | 51.08 |
| | Random | 19.24 | 31.84 | 63.33 | 73.18 | 63.54 | 60.85 | 64.44 | 36.26 | 38.00 | 57.09 |
| | Channel | 49.03 | 106.48 | 62.39 | 66.87 | 49.17 | 58.96 | 49.62 | 31.83 | 33.20 | 50.29 |
| | Block | 16.01 | 29.28 | 67.68 | 77.15 | 73.41 | 65.11 | 68.35 | 38.40 | 42.40 | 61.79 |
| Ratio = 20% w/ tune | L2 | 20.97 | 38.05 | 73.24 | 76.77 | 71.86 | 64.64 | 67.59 | 39.93 | 40.80 | 62.12 |
| | Random | 16.84 | 31.98 | 64.19 | 76.06 | 68.89 | 63.30 | 66.88 | 38.31 | 40.80 | 59.78 |
| | Channel | 17.58 | 29.76 | 69.20 | 76.55 | 68.89 | 66.38 | 62.08 | 38.99 | 39.60 | 60.24 |
| | Block | 15.18 | 28.08 | 70.31 | 77.91 | 75.16 | 67.88 | 71.09 | 42.41 | 43.40 | 64.02 |

**Implementation Details.** In the model pruning process, we use 10 randomly selected samples from Bookcorpus [75], each truncated to a sequence length of 128, as the calibration samples for establishing dependency and calculating the gradient for both LLaMA and Vicuna. For ChatGLM, we select 10 random samples from DailyDialog [27]. During the recovery phase, we utilize the cleaned version of Alpaca [49], which comprises approximately 50k samples. Remarkably, tuning these samples requires merely 3 hours on a single GPU with only 2 epochs. More hyper-parameters of pruning and training can be found in Appendix B.

**Statistics of the Compressed Model.** Table 3 presents the statistic of the 7B models that are used in our experiments: the parameter count, MACs, memory requirements and latency for running each model. The statistical evaluation is conducted using the infer-

Table 3: Statistics of the base model and the compressed model.

| Model | Strategy | Ratio | #Params | #MACs | Memory | Latency |
|---|---|---|---|---|---|---|
| LLaMA-7B Vicuna-7B | - | - | 6.74B | 424.02G | 12884.5MiB | 69.32s |
| | Channel | 20% | 5.39B | 339.36G | 10363.6MiB | 61.50s |
| | Block | 20% | 5.42B | 339.60G | 10375.5MiB | 58.55s |
| | Channel | 50% | 3.37B | 212.58G | 6556.3MiB | 40.11s |
| | Block | 50% | 3.35B | 206.59G | 6533.9MiB | 37.54s |

ence mode, where the model is fed a sentence consisting of 64 tokens. The latency is tested under the test set of WikiText2 on a single A5000. Here, the 'Block' strategy implies that the pruned unit in the model consists of Group Type A and Group Type B as illustrated in Figure 2, whereas 'Channel' indicates that the unit to be pruned is Group Type C. We delve into an analysis of these two choices in Section 4.2(Channel Strategy vs. Block Strategy). The pruning ratio stated here denotes the approximate ratio of parameters to be pruned since the number of parameters within each pruned structure does not perfectly match the total number of pruned parameters.

## 4.2 Zero-shot Performance

Table 1,2,4 and 5 shows the zero-shot performance of the pruned model. Based on the evaluation conducted on LLaMA, employing a 20% parameter reduction without post-training, the pruned

Table 4: Zero-shot performance of the compressed Vicuna-7B

| Pruned Model | Method | WikiText2 ↓ | PTB↓ | BoolQ | PIQA | HellaSwag | WinoGrande | ARC-e | ARC-c | OBQA | Average |
|---|---|---|---|---|---|---|---|---|---|---|---|
| Ratio = 0% | Vicuna-7B | 16.11 | 61.37 | 76.57 | 77.75 | 70.64 | 67.40 | 65.11 | 41.21 | 40.80 | 62.78 |
| Ratio = 20% w/o tune | l2 | 3539.98 | 5882.21 | 55.90 | 56.15 | 32.37 | 51.85 | 30.01 | 28.41 | 28.20 | 40.41 |
| | random | 34.63 | 112.44 | 61.47 | 70.89 | 54.67 | 56.27 | 55.60 | 31.74 | 34.60 | 52.18 |
| | Channel | 71.75 | 198.88 | 51.77 | 63.93 | 42.58 | 55.17 | 43.94 | 29.27 | 33.40 | 45.72 |
| | Vector | 27.03 | 92.51 | 62.17 | 71.44 | 55.80 | 53.43 | 55.77 | 33.28 | 37.80 | 52.81 |
| | Element[2] | 24.70 | 94.34 | 62.87 | 75.41 | 64.00 | 58.41 | 60.98 | 37.12 | 39.00 | 56.83 |
| | Element[1] | 25.74 | 92.88 | 61.70 | 75.30 | 63.75 | 56.20 | 63.22 | 36.60 | 37.00 | 56.25 |
| Ratio = 20% w/ tune | Vector | 19.94 | **74.66** | 63.15 | 74.59 | 61.95 | 60.30 | 60.48 | 36.60 | **39.40** | 56.64 |
| | Element[2] | **18.97** | 76.78 | 60.40 | 75.63 | **65.45** | **63.22** | 63.05 | 37.71 | 39.00 | **57.78** |
| | Element[1] | 19.69 | 78.25 | **63.33** | **76.17** | 65.13 | 60.22 | 62.84 | 37.12 | 39.20 | 57.71 |

model manages to retain 89.8% of the performance exhibited by the unpruned model. Furthermore, through the efficient post-training, the classification accuracy further improves to 60.07%, achieving 94.97% of the accuracy attained by the original model. This demonstration proves the feasibility of using LLM-Pruner to effectively compress the model, even without relying on training data, and within a remarkably short period of time. Surprisingly, we discover that on most datasets, the pruned model with 5.4B LLaMA even outperformed chatGLM-6B. This highlights the superiority of the LLM-Pruner: if a smaller model with a customized size is required, LLM-Pruner is more cost-effective compared to retraining another model with a satisfying performance. However, with 50% parameters pruned, a large accuracy degradation is observed (see Appendix C.4). Compressing LLMs under high compression rates still remains a large challenge.

The compression results of Vicuna-7B align with those of LLaMA, as pruning 20% of parameters on Vicuna-7B maintains performance at 92.03% of the original model. We test a smaller pruning rate of 10% on chatGLM-7B, where the pruned model only experiences a marginal performance decrease of 0.89%, which can be recovered through post-training. Despite the pruned model outperforming the uncompressed model, we don't assert it is better than the original model. This is largely because chatGLM-6B, a bilingual model, has limited English pre-training exposure. Post-training, however, introduces it to more English corpus, albeit limited, improving its English comprehension.

**Ablation: Impact of Importance Estimation.** We conduct tests on all proposed importance estimation techniques mentioned in Section 3.2. The results can be found in Table 1 and 4. Here, *Element*[n] represents the importance evaluation utilizing the n-th order term in Eq.5. *Vector* represents the result corresponding to Eq.3. Based on the results obtained from LLaMA-7B and Vicuna-7B, pruning algorithms achieved the best average performance mostly by leveraging the second-order derivatives for each parameter. Nonetheless, given that first-order derivatives are considerably more efficient than second-order derivatives, though yielding slightly inferior results, we still vote for the first-order term as a competitive method. Besides, the results on chatGLM-7B differed significantly from these findings. The importance estimation on each parameter fails, performing even worse than l2, while the importance estimation on the weight matrix reaches the best performance.

**Ablation: Channel Strategy vs. Block Strategy.** From the results presented in Table 2, it is evident that pruning 'Channel' significantly deteriorates performance compared to pruning 'Block'. This discrepancy arises because the layers within the stacked transformer do not evenly distribute their importance. As shown in Figure 3, the first and last layers have a profound impact on the model's performance, and pruning them results in more substantial performance degradation compared to other layers. However, due to the uniform treatment of the 'Channel' group across all layers, it becomes inevitable to prune the first and last layers, leading to a significant decline in performance.

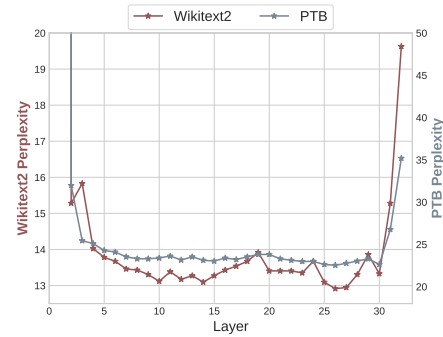

Figure 3: Layer sensitivity for Pruning: Removing Groups in only one layer.

Table 5: Zero-shot Performance of the compressed ChatGLM-6B

| Pruned Model | Method | PIQA | HellaSwag | WinoGrande | ARC-e | ARC-c | OBQA | Average |
|---|---|---|---|---|---|---|---|---|
| Ratio = 0% | ChatGLM-6B | 67.95 | 46.37 | 52.33 | 48.36 | 29.95 | 37.40 | 47.05 |
| Ratio = 10% w/o tune | L2 | 61.97 | 37.22 | 49.72 | 42.05 | 28.24 | 35.40 | 42.43 |
| | Random | 65.29 | 43.18 | 51.30 | 47.52 | 29.52 | 34.60 | 45.24 |
| | Vector | 66.32 | 43.51 | 53.04 | 47.56 | **30.72** | **35.80** | 46.16 |
| | Element[1] | 54.35 | 28.07 | 50.59 | 27.82 | 24.66 | 33.20 | 36.45 |
| w/ tune | Vector | **67.74** | **46.35** | **53.99** | **51.01** | 29.95 | 35.00 | **47.34** |

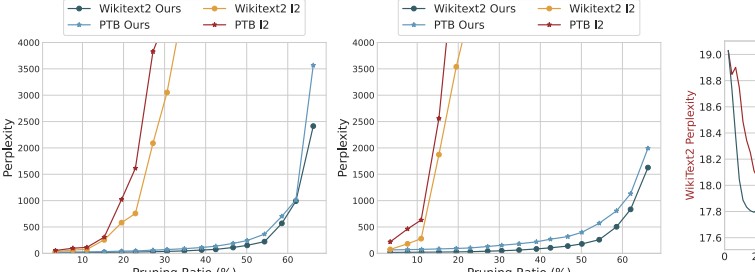

Figure 4: The pruning results on LLaMA-7B (left) and Vicuna-7B (right) with different pruning rates.

Figure 5: Perplexity on zero-shot datasets across varyhing steps.

## 4.3 More Analysis

**Impact of Different Pruning Rates.** We investigate the impact of pruning the LLM at various pruning ratios in Figure 4. We compare our pruning results with the L2 strategy because L2 is also a data-free pruning algorithm. It is observed in the experiment of LLaMA that when the pruning ratio reaches approximately 20%, the magnitude-dependent algorithm experiences a rapid collapse, leading to the loss of information. Conversely, by employing LLM-Pruner, we are able to increase the pruning ratio to around 60% while achieving an equivalent perplexity level. Furthermore, in the case of Vicuna-7B, removing 10% parameters results in a performance decline equivalent to that of LLM-Pruner with 60%. The utilization of LLM-Pruner enables a significant increase in the number of model parameters that can be pruned, thereby substantially reducing computational overhead.

**Tuning on the External Dataset.** To tune the pruned model, we utilize the external dataset Alpaca [49]. The evaluation curves of the pruned model on two zero-shot datasets during the post-training process are depicted in Figure 5. The results demonstrate a rapid decrease in the perplexity of the pruned model within 300 steps, followed by a gradual increase. We provide a more comprehensive evaluation in Appendix C.3. It is important to note that if the model is trained for an excessive number of steps, it runs the risk of overfitting the external dataset, potentially compromising its performance in other general-purpose tasks.

**Impact of Dependency-based Structured Pruning.** To study the importance of dependency-based structural pruning, we conduct an experiment to disrupt dependencies within groups, where each weight matrix $W_i$ is pruned solely based on the importance score estimated on itself. Table 6 presents the results demonstrating the impact of dependencies in structural pruning. In the absence of dependencies, the model nearly fails in the zero-shot generation and classification tasks. Even with tuning, the model fails to recover, showing a substantial difference compared to the results in dependency-based pruning.

**Impact of Different Aggregation Strategies.** We conduct tests on the aggregation algorithms proposed in Section 3.2. Our experimental results unveil notable discrepancies in model performance across different aggregation strategies, with particular emphasis on the 'Last-only' strategy. Among the evaluated approaches, the 'Max' strategy attains the most favorable outcomes in terms of perplexity, signifying enhanced coherence and fluency in sentence generation. However, it is important to note that the 'Max' strategy exhibits the poorest zero-shot classification results compared to all four strategies. Conversely, the 'Last-only' strategy showcases superior classification performance but

Table 6: Effect of the dependency-based structural pruning. Average represents the average performance on 7 classification datasets.

| | Method | WikiText2↓ | PTB↓ | Average↑ |
|---|---|---|---|---|
| w/o Tuning | w/o dependency | 68378.42 | 79942.47 | 38.32 |
| | w/ dependency | 19.09 | 34.21 | 56.69 |
| w/ Tuning | w/o dependency | 13307.46 | 13548.08 | 38.10 |
| | w/ dependency | 17.58 | 30.11 | 59.23 |

Table 7: Impact of different aggregation strategies on group importance estimation. Experiments are performed on LLaMA-7B.

| Method | WikiText2↓ | PTB↓ | ARC-e↑ | PIQA↑ | OBQA↑ |
|---|---|---|---|---|---|
| Summation | 66.13 | 164.25 | 40.70 | 63.49 | 34.80 |
| Max | 62.59 | 144.38 | 39.60 | 63.71 | 34.60 |
| Production | 77.63 | 192.88 | 37.84 | 62.08 | 35.00 |
| Last-only | 130.00 | 170.88 | 41.92 | 64.75 | 35.20 |

suffers from the poorest generation quality. In our experiments, we make a trade-off by selecting the 'Summation' strategy since it shows both good generalization quality and classification performance.

**Comparison with DistilBERT**   We show the comparison results of DistilBERT and LLM-Pruner on LLaMA-7B in Table 8. LLM-Pruner outperforms DistilBERT by 4.24% on average with even a smaller size. The reason lies in that LLM-Pruner minimizes model disruption during pruning, whereas Distil-BERT merely selects one layer out of two. As a result, the model pruned by LLM-Pruner demands less data to recover its performance compared with DistilBERT, consequently achieving superior performance.

Table 8: DistilBert vs. LLM-Pruner. The average here means the average score on the above seven datasets.

| Pruning Ratio | #Param | Average |
|---|---|---|
| DistilBert | 3.50B | 44.64 |
| LLM-Pruner | 3.35B | 48.88 |

**Scratch Training vs. Pruning.**   We compare LLM-Pruner with StableLM-3B[4] with a similar parameter size. To ensure fairness, both models are fine-tuned on the Alpaca dataset. The experimental results of these two models are shown in the Table 9. LLM-Pruner crafts lightweight LLMs with low resources, and even can sometimes achieve better performance than LLMs from scratch training. However, we also acknowledge that the LLaMA-3B obtained by LLM-Pruner will not always outperform other 3B models from scratch training, due to the huge gap in the size of training corpus.

Table 9: Scratch Training (StableLM-3B) vs. Pruning (LLaMA-3B by LLM-Pruner)

| Pruning Ratio | #Param | Latency | BoolQ | PIQA | HellaSwag | WinoGrande | ARC-e | ARC-c | OBQA | Average |
|---|---|---|---|---|---|---|---|---|---|---|
| StableLM-3B | 3.6B | 31.69s | 48.78 | 69.48 | 44.52 | 54.62 | 50.93 | 25.17 | 27.40 | 45.84 |
| LLaMA-3B | 3.6B | 37.96s | 61.41 | 70.08 | 51.01 | 55.01 | 46.80 | 30.38 | 37.40 | 50.30 |

**More Data for Recovery**   Despite our primary experiments being conducted using 50k samples, we remain convinced that the inclusion of additional data could substantially enhance the recovery process, albeit at a considerably higher computational cost. Consequently, we conduct an experiment aimed at model recovery with more data, employing a dataset comprising 2.59 million samples [59]. The results are detailed in Table 10. From the results, it is evident that the performance of the compressed model closely approximates that of the base model, exhibiting only a marginal performance decrease of 0.89%.

Table 10: Model Recovery: 50k samples vs. 2.59M samples

| Model | #Samples | BoolQ | PIQA | HellaSwag | WinoGrande | ARC-e | ARC-c | OBQA | Average |
|---|---|---|---|---|---|---|---|---|---|
| LLaMA-7B | - | 73.18 | 78.35 | 72.99 | 67.01 | 67.45 | 41.38 | 42.40 | 63.25 |
| LLaMA-5.4B | 50k [49] | 64.62 | 77.20 | 68.80 | 63.14 | 64.31 | 36.77 | 39.80 | 59.23 |
| LLaMA-5.4B | 2.59M [59] | 76.57 | 77.37 | 66.60 | 65.82 | 70.62 | 40.70 | 38.80 | 62.36 |

**Case Study.**   We provide some examples of sentences generated by the model compressed using LLM-Pruner in Table 11. We made efforts to ensure a minimal overlap between these generated sentences and the information contained in the tuning corpus, which demonstrates that the information originates from the original model rather than the tuning corpus. We provide additional examples in the Appendix, including the generated sentences of the model without post-training. From the

---

[4]https://huggingface.co/stabilityai/stablelm-tuned-alpha-3b

Table 11: Some examples of generations obtained with the original model and the pruned model.

| Model | Original Model | Pruned Model |
|---|---|---|
| LLaMa | The universe is the entirety of space, time, matter, and energy that exists. It includes all matter and energy, whether visible or not, and in fact the whole universe is invisible. There are many theories regarding its origin and ultimate fate... | The universe is the entirety of space, time, matter, and energy that exists. The laws that make up physics — electrons orbiting atoms, magnets attracting other objects, planets orbiting the Sun—have been consistent over ... |
| Vicuna | 10 steps to build an ios app: 1. Research and develop the idea. 2. Decide on the platform you want to use (iOS, Android, or both). 3. Create a plan and budget. 4. Hire a team of developers. 5. Design the user interface (UI). 6. Develop the user interface (UI). 7. Test and iterate on the app. 8. Submit the app to the app store. 9. Optimize the app for performance and user experience (UX). 10. Market and promote the app. | 10 steps to build an ios app: 1. Plan your app: decide what it should do, what it will look like, and what it will cost to build it. 2. Hire a developer: A good iOS developer will help you design the user interface, create the app, and make sure it's bug-free. 3. Create a user interface: This is where the magic happens. Your app's user interface will be designed and mocked up at this point... |
| ChatGLM | Write a poetry about love and peace. In the midst of life's hustle and bustle, I find your love a soothing peace. It is a feeling that cannot be conveyed, A connection that cannot be distanced. With your smile, your eyes, your embrace, I feel a sense of belonging that only you can give. It is a love that lasts forever, A love that cannot be defeat... | Write a poetry about love and peace. Love and peace, two such wonderful things\ A pure and endless desire in my heart\ And both of them, I must seek for\ A long, long time, I know..\ Love, I know, is a feeling of being\ A perfect partner, in every sense\ And peace, I need it, so much, one day\ A long, long way, my heart will go.. |

cases in Table 11, it is evident that the sentences generated by the compressed model are comparable to those produced by the original model. They exhibit fluency, relevance, and informativeness regarding the given topic. Nevertheless, during our experiments, we observed that the pruned model's performance deviates from that of the original model, particularly when generating lengthy sentences. Occasionally, it may generate sentences that are meaningless or contain repetitive tokens.

## 5 Conclusion

In this paper, we propose LLM-Pruner, a structured pruning approach for large language models. LLM-Pruner aims to compress sizable language models in a task-agnostic manner while minimizing the dependency on the original training corpus and preserving the linguistic capabilities of LLMs. LLM-Pruner accomplishes this by iteratively examining each neuron within the model as a trigger for identifying dependency groups, thereby constructing the LLM's dependency graph. Subsequently, LLM-Pruner assesses the importance of these groups using both parameter-wise and weight-wise estimation. Finally, we utilize LoRA for fast recovery and adjustment of the pruned model. We evaluate the efficacy of LLM-Pruner on three distinct models—LLaMA, Vicuna, and ChatGLM—utilizing various zero-shot datasets. Our experimental results indicate that LLM-Pruner successfully prunes the model, reducing computational burden while retaining its zero-shot capabilities. Nevertheless, considerable performance degradation occurs when employing high pruning rates, such as the removal of 50% of LLaMA's parameters, resulting in a substantial decline in model performance. Additionally, we observe instances in which the model generates incoherent sentences. Addressing the challenges associated with compressing LLMs at higher pruning rates remains a challenging task.

## Acknowledgment

This project is supported by the Ministry of Education, Singapore, under its Academic Research Fund Tier 2 (Award Number: MOE-T2EP20122-0006), and the National Research Foundation, Singapore, under its Medium Sized Center for Advanced Robotics Technology Innovation.

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

# A    Detailed Explanations for the Dependency Rules

**Group B: Multi-head Attention**

Case 1: Node J is dependent on Node I    Case 2: Node I is dependent on Node J    Case 3: Node K is not dependent on Node J

Figure 6: Illustrations of the two dependency rules. All the cases are extracted from the multi-head attention module.

We provide a detailed explanation of the two dependency rules. It is important to note that these dependency rules do not pertain solely to the forward computation. Instead, they represent directional relationships that exist in both directions. For instance, removing a node in a subsequent layer may also result in the pruning of a node in the preceding layer. Recall the two dependency rules as follows:

$$N_j \in \text{Out}(N_i) \wedge \text{Deg}^-(N_j) = 1 \Rightarrow N_j \text{ is dependent on } N_i \tag{7}$$

$$N_i \in \text{In}(N_j) \wedge \text{Deg}^+(N_i) = 1 \Rightarrow N_i \text{ is dependent on } N_j \tag{8}$$

where $N_i$ and $N_j$ are two neurons. $\text{In}(N_i)$ and $\text{Out}(N_i)$ represents all the neurons that point towards or point from $N_i$. $\text{Deg}^-(N_i)$ and $\text{Deg}^+(N_i)$ represents the in-degree and out-degree of neuron $N_i$. Figure 6 serves as an illustration of the two dependency rules:

- In case 1, Node I and Node J satisfy the rule stated in Eq.7. Consequently, Node J depends on Node I. When Node I is pruned, it is necessary to prune Node J as well.

- In case 2, Node I and Node J satisfy the rule Eq.8. Thus, Node I is dependent on Node J. If Node J is pruned, it becomes imperative to prune Node I as well.

- In case 3, Node J and Node K do not meet the requirement of Eq.7 due to the mismatch in $\text{Deg}^-(N_k) \neq 1$. Thus, with Node J pruned, Node K would not be affected.

# B    Implementation Details

## B.1    For Pruning

**For the baseline**    Given the lack of previous work on the structural pruning of Large Language Models in a task-agnostic and low-resource setting, there is currently no existing baseline for our model. To provide a comprehensive demonstration of the effectiveness of LLM-Pruner, we employ two additional methods for evaluation, alongside the data-free pruning method. All of these methods are built upon the dependent groups identified in Section 3.1:

- L2: We assess the importance of each group based on the magnitude of its weight matrix.

- Random: This method involves randomly selecting certain groups for pruning.

**For the 'Block' Group.**    Based on the findings presented in Table 3, it is preferable to leave the first three layers and the final layer unchanged, as modifying parameters in those layers significantly impacts the model. Within each module, such as the MLP or the Multi-head Attention, the discovered groups are pruned based on a pre-set ratio. For instance, in the MLP layer of LLaMA-7B, we identified 11,008 groups, and with a 25% pruning ratio, the module would prune 2,752 groups. It is worth noting that the pruning rate for the selected groups is higher than the pruning ratio for the parameters, as certain layers (e.g., the embedding layer and excluded layers mentioned) retain their parameters. When aiming for a parameter pruning ratio of 20%, we prune 25% from Layer 5 to Layer 30. Similarly, for a 50% parameter removal, we prune 60% of the groups from Layer 4 to Layer 30.

**For the 'Channel' Group.** The Group 'Channel' exhibits a resemblance to dimension pruning in the model, targeting the pruning of certain dimensions. In the case of the Query, Key, and Value projection in MHA, only the input dimension is pruned, while for the Output projection in MHA, only the output dimension is pruned. It is important to note that the entire dependency is established automatically, without any manual design involved. The 'Channel' group operates in a complementary manner to the 'Block Group'. In the 'Channel' Group, the pruning ratio of the group equals to the pruning ratio of the parameters, as all weight matrices, including the embedding matrix, undergo pruning. Therefore, a 20% pruning ratio of parameters implies pruning 20% of the groups, while a 50% pruning ratio implies pruning 50% of the groups.

### B.2 For Recovery Stage

We follow [19] in our recovery stage. We set the rank $d$ to 8 in our experiment. The learning rate is set to 1e-4 with 100 warming steps. The batch size of training is selected from {64, 128} and the AdamW optimizer is employed in our experiment. The best training epoch we found is 2 epochs, as training with more epochs even has a negative impact on the model performance. We run our experiment on a single GPU with 24GB memory, using approximately 2.5 hours if RTX4090 is utilized. All the linear module is taken into account for efficient tuning. An ablation experiment for this is shown in Table 12.

Table 12: Ablation: Tuning different modules in the recovery stage

| Module | WikiText $\downarrow$ | PTB $\downarrow$ |
|---|---|---|
| ALL | 17.36 | 29.99 |
| - MLP | 17.64 | 30.63 |
| - MHA | 17.62 | 30.23 |

## C More Analysis

### C.1 Pruning vs. Quantization

Here, we conduct a comparative analysis of different compression techniques and illustrate that these techniques can be effectively combined with little performance degradation. We have chosen LLM.int8() [8] as a representative example of quantization methods. Our results show that LLM.int8() outperforms LLM-Pruner while LLM-Pruner enhances latency, reduces parameter size. When these two techniques are applied in tandem, they collectively reduce memory consumption and expedite inference, offering a balanced approach that combines the benefits of both methods.

Table 13: Pruning and Quantization on LLaMA-7B

| Pruning Ratio | #Param | Memory | Latency | BoolQ | PIQA | HellaSwag | WinoGrande | ARC-e | ARC-c | OBQA | Average |
|---|---|---|---|---|---|---|---|---|---|---|---|
| LLaMA-7B | 6.74B | 12884.5MiB | 69.32s | 73.18 | 78.35 | 72.99 | 67.01 | 67.45 | 41.38 | 42.40 | 63.25 |
| LLM.int8() | 6.74B | 6777.7MiB | 76.20s | 73.36 | 78.18 | 73.01 | 66.93 | 67.47 | 40.87 | 41.80 | 63.09 |
| LLaMA-5.4B | 5.47B | 10488.4MiB | 58.55s | 76.57 | 77.37 | 66.60 | 65.82 | 70.62 | 40.70 | 38.80 | 62.36 |
| LLaMA-5.4B + LLM.int8() | 5.47B | 5444.37MiB | 63.10s | 76.39 | 76.71 | 66.62 | 66.46 | 70.54 | 40.19 | 39.20 | 62.30 |

### C.2 Global Pruning vs. Local Pruning

we present a comparative analysis between global pruning and local pruning, where the pruning ratio is 20% and the base model is LLaMA-7B. Global pruning refers to ranking all groups in the model together, whereas local pruning involves only ranking groups within the same module for pruning. The outcome of global pruning leads to varying widths across different layers and modules, whereas local pruning ensures uniformity across all layers.

Based on our experimental findings, we observed a slight advantage of local pruning over global pruning. We think this is because of the varying magnitudes in different layers or modules, which makes the importance scores incomparable between groups across different layers.

Table 14: Results of global pruning and local pruning

| Method | WikiText2 $\downarrow$ | PTB $\downarrow$ | BoolQ | PIQA | HellaSwag | WinoGrande | ARC-e | ARC-c | OBQA | Average |
|---|---|---|---|---|---|---|---|---|---|---|
| Element[1] - local | 19.09 | 34.21 | 57.06 | 75.68 | 66.80 | 59.83 | 60.94 | 36.52 | 40.00 | 56.69 |
| Element[1] - global | 20.84 | 32.86 | 63.15 | 73.23 | 63.31 | 66.38 | 55.85 | 35.49 | 38.00 | 56.49 |

## C.3 Overfitting Phenomena in Post-Training

We present a comprehensive analysis of the overfitting issue in the recovery stage, as previously mentioned in Figure 5. Here the results cover all 9 datasets across various training steps. Based on the findings presented in Table 15, a noticeable trend emerges: the accuracy or generation quality initially shows improvement but subsequently experiences a slight decline. This pattern suggests that the recovery process is completed within a short period. And given that the training corpus is domain-constrained, more training epochs can result in overfitting to the specific dataset while potentially compromising the original capabilities of the language model.

Table 15: The PPL and Accuracy on different training steps

| Step | WikiText2↓ | PTB↓ | BoolQ | PIQA | HellaSwag | WinoGrande | ARC-e | ARC-c | OBQA | Average |
|---|---|---|---|---|---|---|---|---|---|---|
| 0 | 19.09 | 34.21 | 57.06 | 75.68 | 66.80 | 59.83 | 60.94 | 36.52 | 40.00 | 56.69 |
| 200 | 18.10 | 30.66 | 64.62 | 77.20 | 68.80 | 63.14 | 64.31 | 36.77 | 39.80 | 59.24 |
| 400 | 17.69 | 30.26 | 63.00 | 76.66 | 68.75 | 63.54 | 64.39 | 37.20 | 40.60 | 59.16 |
| 600 | 17.69 | 30.57 | 66.24 | 76.28 | 68.52 | 63.85 | 64.48 | 37.37 | 41.00 | 59.68 |
| 800 | 17.64 | 30.57 | 65.05 | 76.22 | 68.38 | 63.77 | 63.64 | 37.29 | 40.80 | 59.31 |
| 1000 | 17.67 | 30.60 | 66.39 | 76.17 | 68.24 | 64.17 | 63.05 | 37.37 | 41.60 | 59.57 |
| 1200 | 17.74 | 30.75 | 65.75 | 76.28 | 68.28 | 63.77 | 63.30 | 37.63 | 41.20 | 59.46 |
| 1400 | 17.88 | 30.85 | 64.34 | 76.28 | 68.31 | 63.85 | 63.47 | 37.80 | 41.20 | 59.32 |

## C.4 Pruning with Large Rates

Additionally, we conducted tests on LLaMA-7B and Vicuna-7B with 50% parameters pruned. We observe a significant decrease in performance compared to the base model. However, the recovery stage proved to be beneficial, resulting in an improvement of approximately 7.39%. Pruning a Language Model with such a high pruning rate remains a challenging task.

Table 16: The PPL and Accuracy on LLaMA-7B with 50% parameters pruned

| Pruned Model | Method | WikiText2↓ | PTB↓ | BoolQ | PIQA | HellaSwag | WinoGrande | ARC-e | ARC-c | OBQA | Average |
|---|---|---|---|---|---|---|---|---|---|---|---|
| Ratio = 50% w/o tune | l2 | 39266.42 | 48867.85 | 55.11 | 53.59 | 27.03 | 49.49 | 26.43 | 29.01 | 34.40 | 39.29 |
| | random | 3887.90 | 4337.27 | 46.79 | 53.37 | 27.50 | 50.59 | 28.07 | 27.90 | 30.00 | 37.75 |
| | Channel | 13891.92 | 16114.91 | 40.37 | 52.18 | 25.72 | 48.86 | 25.72 | 28.24 | 30.40 | 35.93 |
| | Vector | 141.06 | 236.24 | 62.17 | 55.11 | 27.25 | 49.88 | 29.00 | 25.77 | 34.00 | 40.45 |
| | Element[2] | 106.07 | 266.65 | 52.57 | 60.45 | 35.86 | 49.01 | 32.83 | 25.51 | 34.80 | 41.58 |
| | Element[1] | 112.44 | 255.38 | 52.32 | 59.63 | 35.64 | 53.20 | 33.50 | 27.22 | 33.40 | 42.13 |
| Ratio = 50% w/ tune | Channel | 1122.15 | 1092.26 | 40.76 | 54.84 | 26.94 | 49.41 | 27.86 | 25.43 | 32.20 | 36.77 |
| | Vector | 43.47 | 68.51 | 62.11 | 64.96 | 40.52 | 51.54 | 46.38 | 28.33 | 32.40 | 46.61 |
| | Element[2] | 45.70 | 69.33 | 61.47 | 68.82 | 47.56 | 55.09 | 46.46 | 28.24 | 35.20 | 48.98 |
| | Element[1] | 38.12 | 66.35 | 60.28 | 69.31 | 47.06 | 53.43 | 45.96 | 29.18 | 35.60 | 48.69 |

Table 17: The PPL and Accuracy on Vicuna-7B with 50% parameters pruned

| Pruned Model | Method | WikiText2↓ | PTB↓ | BoolQ | PIQA | HellaSwag | WinoGrande | ARC-e | ARC-c | OBQA | Average |
|---|---|---|---|---|---|---|---|---|---|---|---|
| Ratio = 0% | Vicuna-7B | 16.11 | 61.37 | 76.57 | 77.75 | 70.64 | 67.40 | 65.11 | 41.21 | 40.80 | 62.78 |
| Ratio = 50% w/o tune | l2 | 54516.03 | 66274.63 | 45.99 | 53.48 | 26.55 | 47.83 | 27.53 | 28.58 | 30.40 | 37.19 |
| | random | 17020.73 | 13676.54 | 48.17 | 53.43 | 27.31 | 50.43 | 26.30 | 29.78 | 30.20 | 37.95 |
| | Channel | 8360.30 | 10651.30 | 38.69 | 53.10 | 26.42 | 50.20 | 25.97 | 29.52 | 29.60 | 36.22 |
| | Vector | 189.87 | 409.75 | 62.14 | 55.33 | 26.99 | 51.93 | 27.86 | 26.02 | 32.60 | 40.41 |
| | Element[2] | 143.85 | 427.77 | 53.76 | 59.79 | 34.86 | 50.28 | 33.29 | 27.30 | 34.60 | 41.98 |
| | Element[1] | 162.38 | 469.74 | 52.45 | 58.65 | 34.27 | 49.72 | 32.41 | 28.75 | 35.80 | 41.72 |
| Ratio = 50% w/ tune | Channel | 206.23 | 526.89 | 62.11 | 58.38 | 32.29 | 51.22 | 33.08 | 24.06 | 30.20 | 41.62 |
| | Vector | 46.11 | 147.52 | 62.14 | 64.91 | 39.80 | 50.91 | 47.77 | 27.30 | 32.60 | 46.49 |
| | Element[2] | 42.99 | 139.91 | 58.87 | 69.48 | 46.38 | 54.78 | 46.89 | 29.01 | 34.80 | 48.60 |
| | Element[1] | 41.64 | 143.74 | 62.08 | 69.48 | 46.79 | 54.54 | 48.15 | 29.78 | 36.20 | 49.57 |

# D Generations From Compressed Model

Table 18, 19, 20 and 21 show more examples of the models pruned by LLM-Pruner. We present the generation results of both the pruned model with post-training and without post-training. The absence of post-training allows us to better understand the information retained in the model. We include the results of ChatGLM-6B in two languages as it is a bilingual model.

Table 18: Generated Examples from the Compressed LLaMA-5.4B

| Model | Generated Sentences |
|---|---|
| LLaMA w/ Tune | Neural Sequential Model, especially transformers, have become very popular due to their impressive performance in different areas. The neural network model can recognize the human voice more efficiently than speech recognition, and speech can be better distinguished. However, its high energy consumption has been the biggest problem. Many efforts have been made, such as minimizing the number of layers and the length of the neural network, but it has had no effect. The work is to develop a low-dimensional sound model which is suitable for the high frequency band of the speech. After reducing the ... |
| | AI can create a logo in seconds or design a book cover, and they need to do it much faster than us humans can. So what are the implications that this has for branding design? Giving Creative Control to Algorithms. The first wave of 'creative' branding has come from designers and agencies submitting their creations to algorithms. They are algorithms that have been programmed to create, and are being asked to replace designers in the creative process. For example, in 1998 Google used a random process to select words and colours for their logo. Now, in |
| | What's great about the holiday season, is it provides us with the excuse we need to reconnect with the ones we have grown apart from. There are many people who I only talk to a couple of times per year during the year and never interact with them anymore, and while I'm completely okay with that – I feel they too should use Christmas as a great opportunity to get back in touch with people in their lives that maybe they were too busy to talk to before. If you are on good terms with a family member or friend, and you haven't been talking to them lately due to how |
| LLaMA w/o Tune | Neural Sequential Model, especially transformers, we designed them as a neural network that transforms input words to corresponding output. In our model, the 73 neural nodes are arranged in the form of L-shaped and are used to construct the two-dimensional lattice and form a sentence. This sentence is then transferred to the decoder to learn the words. Through 5.4.1 million repetitions, we can form a language model through the neural network we designed. With the help of a word-level data correction method, we made a strong progress in the recognition of human language. |
| | AI can create a logo in seconds. You'll see a design in an instant and you'll be ?? that you're getting one the best logo designs around. The process is quick and easy and if you need it now you'll pay just a fraction of the cost for a professional logo designer. Why would I need a logo? It helps a brand stand out from the competition. It's the visual cue that people respond to - the logo. It's what makes you stop and look at the brand when you're walking by a street. Why invest in a logo? |
| | What's great about the holiday season is being with the people that you love, and the holidays are the perfect occasion to celebrate with the people that you love most. Being together with family and friends is one of those special occasions that can have a remarkable effect on how we perceive our lives, which is why they become some of the most cherished memories of our lifetimes. But holiday planning is more complex than a simple gathering. In fact, many families need professional advice to find the right way to handle the emotions involved. And that's why planning, preparation, |

Table 19: Generated Examples from the Compressed Vicuna-5.4B

| Model | Generated Sentences |
|---|---|
| Vicuna w/ Tune | Last night we all danced together in the rain, all our cares and worries forgotten. And as the sun began to rise, I looked into your eyes and knew that everything was going to be okay. That's the way it is now, forever and forever. I love you, my baby, and I always will. |
| | It fills me with such pride and joy to be working along side her, learning from her and sharing our experiences, joys, and challenges as partners. Our bond is growing stronger each day and I can't help but feel grateful for the beautiful life she has brought into my own. To my loving wife, I love you more and more each day. I can't wait to spend the rest of my life with you and raising our future children together proyecting the light of Christ into the world. And to all of you reading this, I encourage you to take this seriously and consider your relationship with the one |
| | Meet me at midnight tonight at the Bell Tower in town. Bring a torch, and we'll take the treasure off the church's hands, no matter the cost.' With a deep sense of trepidation, Alaric finds himself agreeing to the plan. He spends the day getting everything he can from the village, preparing for the inevitable attack he'll have to launch. When the day arrives, he and his companions mount their steeds and set off towards the towering keep of the temple. As they draw closer, the wind whispers |
| Vicuna w/o Tune | Last night we all danced together in the rain. The water felt good on my face. The song played over and over, but I didn't care. I just kept on dancing with you. I don't know what to say. I'm not good with words. But I know that I've never felt this way before. With anyone. Not even myself. And I don't want to let that feeling go. I don't know what you feel, or what you want from me. But I know that this is real. And I don't want to let it go |
| | It fills me with such pride and joy, it's like seeing my baby for the first time, and I just want to hold it close and never let it go. The thought that it might have to be put down one day breaks my heart. I want to do everything in my power to protect it, to keep it safe and give it the best life possible. But it's not mine. It's never been mine to keep. And I know that one day, sooner or later, it will have to be given away. I can't help but feel a sense of responsibility that I don't ... |
| | Meet me at midnight, I promise you'll be the only one there. It's hard for me to believe, but I'm scared. I don't want to be alone tonight, but I know I have to be brave. Hurry down the hill, I'm waiting for you, I'll take you to a place that I know is safe. We won't be seen, nobody'll know. It's almost time for you to go. I don't want you to be late tonight. |

Table 20: Generated Examples from the Compressed ChatGLM-5.6B (English)

| Model | Generated Sentences |
|---|---|
| ChatGLM w/ Tune | How to buy a cat? It's important to understand that while it's understandable that you might be looking todnle a cat for yourself, adoption is a great way to bring home a furry friend without having to shop. Here are some things to consider before adoption: 1. First, make sure you know you're interested in Jr. or Vioness. This determines whether you're looking for a young adult cat or an older cat. 2. research your options: Look up reputable cat owners or organizations that foster cats in your area. This will give you information on cat owners |
| | What is McDonald's? McDonald's is a popular American fast-food restaurant chain that offers a variety of food and beverage items. The company was founded in 1940 by Richard and Maurice Mc Eu Eu in San Francisco, and the chain was originally called "McNugguggets." Today, McDonald's is known for its popular fast-food items such as McDonald's Drive-Thru, McChickens, and McFlurryrry. The company also offers a wide range of other food and beverage items such as Happy Endings, Big Macs, and ... |
| | What is Mean Square Error? The Mean Square Error (MSE) is a measure of the difference between a target value and the best possible estimate of its true value, based on the data. It is calculated by taking the average difference between the actual value and the estimated value over the※ number of times the target value is repeated. |
| ChatGLM w/o Tune | How to buy a cat? It's important to understand that adopting a cat from a shelter or organization is a valuable and responsible adoption option, as well as one that may be safer and more comfortable for you and your family. Additionally, it's important to read carefully before adopting to ensure that you choose a Cat who is suitable for you and your family. |
| | What is McDonald's? McDonald's is a popular fast-food restaurant chain from United States known for its restaurant concepts such as McNugguggets, McFlurryrry, and McRibibs. The chain started in 1955 as a small operation operation called "Speedy McDonald's" in San Antonio, Texas and it quickly expanded and became a widely recognized and popular phenomenon in the country. Today, McDonald's is global icon known around the world. |
| | What is Mean Square Error? Mean Square Error (MSE) is a measure of the accuracy of a function or a measurement when it is repeated and the result is expected to be within a certain range. It is calculated by taking the difference between two sets of measurements and measuring the difference squared. In simple terms, it means the mean sum of the errors over a given period of time or the difference between the actual and expected values of ... |

Table 21: Generated Examples from the Compressed ChatGLM-5.6B (Chinese)

| Model | Generated Sentences |
|---|---|
| ChatGLM w/ Tune | 请给我推荐两个北京著名的景点 北京著名的景点有很多,以下是两个:1. 颐和园:颐和园是北京的一座皇家园林,也是世界文化遗产。它是一座是一座结合了中国传统建筑和现代艺术的杰作,被誉为"万园之园"。颐和园内有很多建筑和景点,包括长廊、排排坐、佛香阁、万寿山等,非常适合漫步和参观。2. 故宫:故宫是中国古代皇宫,也是世界文化遗产,是北京的必去之地。它是世界上是世界上现存最大的、最完整的、最完整的、最著名的古代宫殿宫殿建筑之一。故宫内有很多建筑和景点,包括紫禁城 |
| | 简单来说,相对论指出 相对论是由爱因斯坦提出的一个关于时间和空间如何解释和变化的的理论,被称为狭义相对论。相对论认为,时间和空间并不是绝对的,而是相对的,也就是说,它们的测量值取决于观察者的惯性参照系。根据相对论,当物体以高速运动或以或以某种方式加速时,时间和空间的测量值会发生变化。最著名的是,时间和空间的测量值会"膨胀",即随着物体运动,它们看起来会变得越来越长或越来越短。 |
| | 如何健康的生活 要保持健康的生活方式,可以参考以下建议:1. 定期锻炼:适量运动可以帮助增强身体健康,减轻压力,改善心情。建议每周至少进行3-5次有氧运动,如慢跑、快走、游泳、瑜伽等。2. 饮食均衡:控制饮食摄入量,摄入摄入适量、健康的蛋白质、碳水化合物、脂肪以及维生素和矿物质。3. 睡眠充足:充足的睡眠有助于恢复身体和大脑功能,提高身体免疫力。建议每晚睡眠7-8小时。4. 勤洗手:避免传染病毒。 |
| ChatGLM w/o Tune | 请给我推荐两个北京著名的景点 北京著名的景点太多,以下是两个必去的景点:1. 故宫博物院: 故宫是世界上是世界上保存最完整的、最著名的宫殿建筑之一,也是中国历史文化名城和世界文化遗产。游客可以参观紫禁城紫禁城,欣赏到欣赏到大量的历史文物和艺术品。2. 长城: 长城是中国古代著名的防御工程之一,也是中国历史文化名城和世界文化遗产。游客可以参观长城的各种不同高度和不同用途的客房现代物流设施,还可以在长城上徒步。 |
| | 简单来说,相对论指出 相对论是由爱因斯坦提出的一组理论和概念,包括两个部分:狭义相对论和广义相对论。狭义相对论是描述在高速运动或缩短物体长度时,时间和空间如何弯曲的理论。它指出,当我们测量一个物体的长度或加速运动时,我们的测量结果观测到的是相对一个观察者观察者静止静止的物体长度或速度的尺子。这个理论解释了为什么光速是恒定不变的不变的。广义无昧相对论是描述引力如何加速和改变物体的质量的。它指出,引力是由质量m和引力场引起的,两个物体引力场的中心的引力是 |
| | 如何健康的生活 想要过健康的生活可以参考以下建议: 1. 保持规律的规律的充足的睡眠:尽可能每晚每晚睡7-8个小时,保持规律的规律的睡眠时间有助于维持身体健康和精神状态。2. 多运动:每天进行适量的运动有助于减轻压力和增强身体免疫力,建议每天进行适量的有氧运动和力量训练。3. 饮食均衡:保持均衡的饮食,包括摄入足够的各种营养素和禁止过量摄入垃圾食品和饮料。4. 减少不习惯:尽可能减少不良习惯,如吸烟、酗酒和过度使用电子设备等。 |

