# OpenReview forum: "LLM-Pruner: On the Structural Pruning of Large Language Models"
_NeurIPS.cc/2023/Conference — NeurIPS 2023 poster_

### Official Review · Reviewer_AFsw · 2023-07-04

**Soundness:** 3 good
**Presentation:** 3 good
**Contribution:** 3 good
**Rating:** 7
**Confidence:** 3

**Summary:**

This paper tackles the problem of reducing the parameter number of large language models with pruning.
It identifies structures within an LLM and uses them to compose a dependency graph-based.
The authors provide experiments where they prune the parameters of LLMS and evaluate them on zero-shoot tasks.
Namely, they consider LLaMA (7B), VICUNA (7B), and ChatGLM (6B).
The results show that pruning can reduce 20% of the parameters and retain 90-94% of the original performance.

**Strengths:**

- This paper addresses a crucial aspect of a more efficient language model - especially when models get enormous.
- It employs efficient methods and evaluates the pruned models on a broad set of tasks.
- The results show that more performance can be preserved than parameters removed during pruning.

**Weaknesses:**

- As the authors show, pruning effectively reduces the parameter of LLMs. But how does the pruned model compare to the next smaller model variation? Showing that with the proposed method, we can get a better-pruned model (7B to 3B) compared to the next smaller model (3N) could make a strong point for this paper.
- This paper considers only the same-sized LLMs, but from recent work, we know that emergent capabilities (when we can call them that) emerges with even bigger models. However, with this proposed approach, we are still determining how it scales to bigger models. Do you have more results on bigger models (13B, 20B, 40B) or insights on how efficient your approach is with a growing number of parameters?
- While referring to efficiency gains in terms of GPU hours, the paper does not provide any further information about which GPU is used. Are the 14 GPU days of TinyBERT and the 3 GPU hours of the proposed approach comparable in terms of GPT hardware?


**Questions:**

- How do you compare your approach with a distilling method (like Distilbert), where general competencies (MLM) are actually distilled from bigger models?


**Limitations:**

---

> ### Author Rebuttal · Authors · 2023-08-09
>
> We truly appreciate the reviewer for the constructive comments.
>
> ---
>
> > As the authors show, pruning effectively reduces the parameter of LLMs. But how does the pruned model compare to the next smaller model variation? Showing that with the proposed method, we can get a better-pruned model (7B to 3B) compared to the next smaller model (3N) could make a strong point for this paper.
>
> We thank the reviewer for the constructive suggestion.  We chose a well-known open-source 3B LLM, StableLM-3B, and compared it with a pruned LLaMA model with a similar parameter count. To ensure fairness, both models were fine-tuned on the Alpaca dataset. The experimental results of these two models are shown in the table below. As evident from the table, our model outperforms StableLM-3B. Additionally, we would like to note that the advantage of our approach lies in the ability to achieve a lightweight LLM with minimal overhead (7.3M vs. 800B tokens for training), which removes the necessity of training a full 3B LLM from scratch. We would include these results in the revised version.
>
> | Pruning Ratio | #Param | Latency | #Tokens for Training | BoolQ | PIQA  | HellaSwag | WinoGrande | ARC-e | ARC-c | OBQA  | Average |
> |-------|--------|---------|--------|-------|-------|--------|------|-------|-------|-------|---------|
> | StableLM-3B   | 3.6B   | 31.69s  | 800 Billion         | 48.78 | 69.48 | 44.52     | 54.62      | 50.93 | 25.17 | 27.40 | 45.84   |
> | LLaMA-3B by LLM-Pruner    | 3.6B   | 37.96s  | 7.3 Million            | 61.41 | 70.08 | 51.01     | 55.01      | 46.80 | 30.38 | 37.40 | 50.30   |
>
> ---
>
> > This paper considers only the same-sized LLMs, but from recent work, we know that emergent capabilities (when we can call them that) emerges with even bigger models. However, with this proposed approach, we are still determining how it scales to bigger models. Do you have more results on bigger models (13B, 20B, 40B) or insights on how efficient your approach is with a growing number of parameters?
>
> Thanks for the comment. We show the pruning results on LLaMA-13B in the table below. The training of the pruned LLaMA-13B takes 7 hours on a single A5000 with 24GB memory, while training a pruned LLaMA-7B takes ~4.5 hours on the same device. Most of the experimental results of LLaMA-13B are consistent with the results on LLaMA-7B. Notably, when comparing the outcomes of experiments conducted on 7B and 13B models, a distinction in parameter redundancy becomes very evident. After pruning 20% of the parameters, LLaMA-13B experiences a relatively minor drop of 3.18% in comparison to LLaMA-7B's decline of 5.86%. Following fine-tuning, the 13B model only exhibits a decrease of 0.95% in performance (from 64.97 to 64.02), outperforming the 3.43% decrease observed in the 7B model (from 63.25 to 59.82).
>
> | Pruning Ratio        | Method    | WikiText2 | PTB    | BoolQ | PIQA  | HellaSwag | WinoGrande | ARC-e | ARC-c | OBQA  | Average |
> |----------------------|-----------|-----------|--------|-------|-------|-----------|------------|-------|-------|-------|---------|
> | Ratio = 0%            | LLaMA-13B | 11.58     | 20.24  | 68.47 | 78.89 | 76.24     | 70.09      | 74.58 | 44.54 | 42.00 | 64.97   |
> | Ratio = 20% w/o tune | L2        | 61.15     | 91.43  | 61.50 | 67.57 | 52.90     | 57.54      | 50.13 | 31.14 | 36.80 | 51.08   |
> |                      | Random    | 19.24     | 31.84  | 63.33 | 73.18 | 63.54     | 60.85      | 64.44 | 36.26 | 38.00 | 57.09   |
> |                      | Channel   | 49.03     | 106.48 | 62.39 | 66.87 | 49.17     | 58.96      | 49.62 | 31.83 | 33.20 | 50.29   |
> |                      | Block     | 16.01     | 29.28  | 67.68 | 77.15 | 73.41     | 65.11      | 68.35 | 38.40 | 42.40 | 61.79   |
> | Ratio = 20% w/ tune  | L2        | 20.97     | 38.05  | 73.24 | 76.77 | 71.86     | 64.64      | 67.59 | 39.93 | 40.80 | 62.12   |
> |                      | Random    | 16.84     | 31.98  | 64.19 | 76.06 | 68.89     | 63.30      | 66.88 | 38.31 | 40.80 | 59.78   |
> |                      | Channel   | 17.58     | 29.76  | 69.20 | 76.55 | 68.89     | 66.38      | 62.08 | 38.99 | 39.60 | 60.24   |
> |                      | Block     | 15.18     | 28.08  | 70.31 | 77.91 | 75.16     | 67.88      | 71.09 | 42.41 | 43.40 | 64.02   |
>
> ---
>
> > W3: While referring to efficiency gains in terms of GPU hours, the paper does not provide any further information about which GPU is used. Are the 14 GPU days of TinyBERT and the 3 GPU hours of the proposed approach comparable in terms of GPT hardware?
>
> We apologize for the imprecise expression here. We will use the size of the dataset instead of GPU training duration to compare the training costs of these two methods. The revision is as follows: In the case of TinyBERT, it was trained for 3 epochs on the Wikipedia dataset with 2500M words, whereas our LLM-Pruner needs only 2 epochs of training on the alpaca dataset with only 5M words.
>
> ---
>
> > Q1: How do you compare your approach with a distilling method (like Distilbert), where general competencies (MLM) are actually distilled from bigger models?
>
> Thanks. We show the comparison results of DistilBERT and LLM-Pruner on LLaMA-7B here. LLM-Pruner outperforms DistilBERT by 4.24% on average, with even a smaller size. The reason lies in  that our approach aims to minimize model disruption during pruning, whereas DistilBERT merely selects one layer out of two. As a result, the model pruned by LLM-Pruner demands less data to recover its performance compared with DistilBERT, consequently achieving superior performance.
>
> | Pruning Ratio | #Param | BoolQ | PIQA  | HellaSwag | WinoGrande | ARC-e | ARC-c | OBQA  | Average |
> |---------------|--------|-------|-------|-----------|------------|-------|-------|-------|---------|
> | DistilBert    | 3.50B  | 54.77 | 63.28 | 40.44     | 52.49      | 43.31 | 27.22 | 31.00 | 44.64   |
> | LLM-Pruner    | 3.35B  | 61.56 | 68.72 | 46.62     | 52.64      | 47.94 | 29.27 | 35.40 | 48.88   |

---

> > ### Comment · Reviewer_AFsw · 2023-08-16
> >
> > Thanks a lot for the feedback and the additional results. I adjusted my scores accordingly.

---

> > > ### Author Response · Authors · 2023-08-16
> > >
> > > Dear Reviewer AFsw,
> > >
> > > We sincerely appreciate your time to review our submission and response. We will follow the reviewer's advice and incorporate the above results into the updated version.
> > >
> > > Best regards,
> > > Authors of Submission 2474

---

### Official Review · Reviewer_mPK3 · 2023-07-05

**Soundness:** 3 good
**Presentation:** 3 good
**Contribution:** 2 fair
**Rating:** 4
**Confidence:** 4

**Summary:**

The paper is trying to present an resource efficient task agnostic technique for pruning large language models. The authors tackle two main problems, 1) in a setup where there is no access to the original dataset that was used to train the LLM, and 2) the pruning should be fast.

The paper presents a three step approach to pruning:
1) Discovery -- finding the groups (structures) that will be evaluated for pruning.
2) Estimation -- estimating the impact of removing a particular group on the performance of the model.
3) Recovery -- post pruning training of the model to recover on the lost performance.

**Strengths:**

The paper presents a novel approach to pruning, which is fast, efficient, and agnostic of the downstream task and operates without the availability of the pretraining dataset.

The paper is well written and clearly explains their approach, and performs a good number of ablation studies to provide a good understanding of the impact of inclusion/exclusion of each component of their approach.

The paper evaluates the work on a wide variety of tasks in zero shot setting.

**Weaknesses:**

The paper starts with the problem that deployment & inference are the key challenges faced by large language models, and presents a solution to it. However, no assessment of latency is provided in the paper. If we prune some parameters, the resulting model is sparse. Sparse models often to less theoretical computations, but at times do not benefit much on latency because the inefficiency introduced by sparse computations.

There is no baseline that is taken into consideration. It it possible to get two models of similar architecture, and prune the larger model to the smaller one, and then train both the pruned model, and the small model, on the Alpaca dataset used in the paper to understand the gain that this approach gives when compared to a model that has been directly trained? A latency analysis between these two models will be pretty useful to make a conclusive statement as well in order for it to be useful in production setups.

**Questions:**

Please see the weakness section. I would be curious to look into latency analysis and comparison with a model directly trained on the same corpus.

**Limitations:**

There isn't any discussion around deploying such a sparse model compared to a dense model, especially whether the theoretical guarantees in terms of speedups translate to the same when deploying.

---

> ### Author Rebuttal · Authors · 2023-08-09
>
> We truly appreciate the reviewer for the constructive comments.
>
> ---
>
> > W1: The paper starts with the problem that deployment \& inference are the key challenges faced by large language models, and presents a solution to it. However, no assessment of latency is provided in the paper. If we prune some parameters, the resulting model is sparse. Sparse models often to less theoretical computations, but at times do not benefit much on latency because the inefficiency introduced by sparse computations.
>
> Thanks for the comment. We show in the table below the latency of the pruned model across different pruning ratios, employing diverse grouping strategies. The latencies shown in the table correspond to runtime evaluations on the test set of WikiText2 and PTB. We test the result on A5000 with 24GB memory.
>
> Regarding sparsity, we would like to note that the proposed approach indeed goes beyond the practice of masking certain parameters to be zero, which is generally known as **unstructural pruning**. Our approach, as indicated in the title, focuses on **structural pruning** of LLM, meaning that we completely remove unnecessary connections and neurons alongside their associated weights, resulting in a **smaller** model rather than a **sparse** one. The model pruned by LLM-Pruner yields improved model acceleration without the need for specialized hardware. By removing unnecessary connections and neurons, structural pruning reduces the number of required multiplications and additions during forward-pass computations. This reduction in floating-point operations leads to faster inference time.
>
> | Group   | Ratio | #Params | Memory     | Latency(WikiText2) | Latency(PTB) |
> |---------|-------|---------|------------|---------------|--------------|
> | /      | 0%     | 6.74B   | 12884.5MiB | 69.32s        | 20.47s       |
> | channel | 20\%  | 5.47B   | 10488.4MiB | 61.50s        | 18.30s       |
> | block   | 20\%  | 5.42B   | 10375.5MiB | 58.55s        | 17.55s       |
> | channel | 50\%  | 3.37B   | 6556.3MiB  | 40.11s        | 11.96s       |
> | block   | 50\%  | 3.35B   | 6533.9MiB  | 37.54s        | 11.27s       |
>
> ---
>
> > W2: There is no baseline that is taken into consideration. It it possible to get two models of similar architecture, and prune the larger model to the smaller one, and then train both the pruned model, and the small model, on the Alpaca dataset used in the paper to understand the gain that this approach gives when compared to a model that has been directly trained? A latency analysis between these two models will be pretty useful to make a conclusive statement as well in order for it to be useful in production setups.
>
> We thank the reviewer for the comment. As advised, we added the StableLM-3B as a baseline. Specifically, we employed the `stabilityai/stablelm-tuned-alpha-3b` model, which has a similar number of parameters and is also tuned on Alpaca to ensure the fairness of comparison.
>
> From the perspective of model performance, the pruned LLaMA-3B model using LLM-Pruner outperforms StableLM-3B, incurring only a minor expense (7.3M vs. 800B tokens) to attain this compact LLM. It demonstrates significant advantages over StableLM on multiple datasets. However, due to the fact that the LLaMA model has a deeper depth (32 layers) but smaller width, while StableLM is shallower (16 layers) but wider, StableLM-3B is slightly quicker than LLaMA-3B pruned by LLM-Pruner. The shallow but wide network is easier for parallel processing of matrix calculation, leading to better inference speeds compared to a deep but narrow network, as also discussed in [1].
>
> | Pruning Ratio | #Param | Latency | #Tokens for Training | BoolQ | PIQA  | HellaSwag | WinoGrande | ARC-e | ARC-c | OBQA  | Average |
> |---------------|--------|---------|----------------------|-------|-------|-----------|------------|-------|-------|-------|---------|
> | StableLM-3B   | 3.6B   | 31.69s  | 800 Billion         | 48.78 | 69.48 | 44.52     | 54.62      | 50.93 | 25.17 | 27.40 | 45.84   |
> | LLaMA-3B  by LLM-Pruner   | 3.6B   | 37.96s  | 7.3 Million            | 61.41 | 70.08 | 51.01     | 55.01      | 46.80 | 30.38 | 37.40 | 50.30   |
>
> [1]DistilBERT, a distilled version of BERT: smaller, faster, cheaper and lighter

---

> > ### Author Response · Authors · 2023-08-18
> >
> > **Additional Explanation: LLM-Pruner produces dense (not sparse) and lightweight models**
> >
> > We sincerely thank the reviewer mPK3 again for the comments. We would like to present in a more intuitive manner that LLM-Pruner produces dense and lightweight models with reduced dimensions, rather than being sparse. Below, we provide an example of the architecture change of LLaMA before and after pruning:
> >
> > After Pruning: (only show one layer for simplicity)
> > ```
> >  LlamaDecoderLayer(
> >         (self_attn): LlamaAttention(
> >           (q_proj): Linear(in_features=4096, out_features=3072, bias=False)
> >           (k_proj): Linear(in_features=4096, out_features=3072, bias=False)
> >           (v_proj): Linear(in_features=4096, out_features=3072, bias=False)
> >           (o_proj): Linear(in_features=3072, out_features=4096, bias=False)
> >         )
> >         (mlp): LlamaMLP(
> >           (gate_proj): Linear(in_features=4096, out_features=8256, bias=False)
> >           (down_proj): Linear(in_features=8256, out_features=4096, bias=False)
> >           (up_proj): Linear(in_features=4096, out_features=8256, bias=False)
> >         )
> >        ...
> > )
> > ```
> >
> > Before Pruning:
> > ```
> > LlamaDecoderLayer(
> >         (self_attn): LlamaAttention(
> >           (q_proj): Linear(in_features=4096, out_features=4096, bias=False)
> >           (k_proj): Linear(in_features=4096, out_features=4096, bias=False)
> >           (v_proj): Linear(in_features=4096, out_features=4096, bias=False)
> >           (o_proj): Linear(in_features=4096, out_features=4096, bias=False)
> >         )
> >         (mlp): LlamaMLP(
> >           (gate_proj): Linear(in_features=4096, out_features=11008, bias=False)
> >           (down_proj): Linear(in_features=11008, out_features=4096, bias=False)
> >           (up_proj): Linear(in_features=4096, out_features=11008, bias=False)
> >         )
> >        ...
> > )
> > ```
> > We would like to hear your further opinion about this, which is essential to us to improve the work. Thank you!
> >
> > Best,
> > Authors of Submission 2474

---

> > > ### Author Response · Authors · 2023-08-21
> > >
> > > Dear Reviewer mPK3:
> > >
> > > As the end of the discussion period approaches, we sincerely look forward to your feedback on our rebuttal. Your insightful comments on our work are really essential and we genuinely aspire to your feedback.
> > >
> > > In response to your concerns about our work, we present the quantitative experiments as follows:
> > > 1. No assessment of latency & No theoretical guarantee of sparse models in terms of speedups -> We provide the latency analysis of the pruned model, and we also show that the pruned model retains its dense structure rather than transforming into a sparse model with sparse computation.
> > > 2. Comparing a smaller model and a larger model with pruning to reduce its size: We conduct a comparison between a LLaMA-3B pruned using LLM-Pruner and an open-source 3B model, StableLM-3B.
> > >
> > > We truly appreciate your time and feedback on our rebuttal.
> > >
> > > Best Regards,
> > > Authors of submission 2474

---

### Official Review · Reviewer_VdKx · 2023-07-07

**Soundness:** 3 good
**Presentation:** 4 excellent
**Contribution:** 3 good
**Rating:** 7
**Confidence:** 3

**Summary:**

The authors propose LLM-Pruner to compress large language models in a task-agnostic manner. Their method can achieve a fast recovery by tuning LoRA after pruning the model. Their experiment results show LLM-Pruner can achieve comparable performance with much smaller memory.

**Strengths:**

1. Their motivation is interesting and the proposed method can be generalized to variable tasks. Their method design also makes sense.
2. Their experiments are extensive, and their analyses also demonstrate the effectiveness of LLM-Pruner.


**Weaknesses:**

1. What's the performance if no post-tuning is applied?
2. What is the performance of other compressing methods like quantization and task-specific pruning?

**Questions:**

None

---

> ### Author Rebuttal · Authors · 2023-08-09
>
> We truly appreciate the reviewer for the constructive comments.
>
> ---
>
> > W1: What's the performance if no post-tuning is applied?
>
> Thanks for the comment. In Table 1-3 in our submitted manuscript, we present the results without post-training (please refer to `Ratio=X% w/o tune`). Without post-training, the models can retain a large portion of their performance with 20% of parameters pruned. For instance, on LLaMA-7B, the model retains 90.7% of the original performance. From the qualitative experimental results of the generated results (see Tables 12-15 in the Appendix), the pruned model with no post-tuning also exhibits decent text generation capabilities.
>
> ---
>
> > W2: What is the performance of other compressing methods like quantization and task-specific pruning?
>
> We thank the reviewer for the comment.
>
> * Task-specific Pruning:
>
> We conducted task-specific pruning on the ARC-easy dataset. In this experiment, both the pruning and tuning process utilized the ARC-easy dataset. The table below shows our experimental results. It can be observed that, compared to LLM-Pruner, there is an improvement in performance on ARC-easy. However, since task-specific pruning is optimized for specific datasets, if we were to perform tests on other datasets, such as BoolQ, the performance would be significantly inferior to that of LLM-Pruner (56.69 vs. 66.79), even without tuning (56.69 vs. 62.54). We believe this is primarily due to task-specific pruning being overfitted to specific domains, thereby not ensuring optimal performance on out-of-domain or zero-shot datasets.
>
> | Method                                                        | In-domain: ARC-easy | Out-of-domain: BoolQ |
> |---------------------------------------------------------------|---------------------|----------------------|
> | Task-specific Pruning + Task-specific Fine-tuning             | 68.35               | 56.69                |
> | LLM-Pruner: Task-agnostic Pruning                             | 61.57               | 62.54                |
> | LLM-Pruner: Task-agnostic Pruning + Task-agnostic Fine-tuning | 64.06               | 66.79                |
>
> * Quantization
>
> We thank the reviewer for pointing out network quantization. By nature, quantization is indeed another line of model compression strategy, which is orthogonal to pruning and hence can be readily deployed on top of pruning to further reduce the network size. The reasons behind lie in that, pruning aims to reduce the number of parameters, while quantization strives for reducing the precision of the numerical values used to represent the parameters. In practice, these techniques are often used in combination to achieve even greater efficiency gains. For example, one can prune a neural network and then quantize the remaining weights and activations to further optimize the model.
>
> We evaluate our techniques against LLM.int8()[1] on LLaMA-7B. We post-train the pruned model obtained by LLM-Pruner under `MBZUAI/LaMini-instruction`. From the results, both quantization and pruning have their own pros and cons: LLM.int8() performs slightly better than LLM-Pruner but LLM-Pruner is faster than LLM.int8() (the speedup ratio of LLM.int() on a 6.7B model is 0.86x, as mentioned in the paper [1]). The pruning of LLMs is still in its early stage,  and there is much that remains to be thoroughly investigated and studied.
>
> [1] LLM.int8(): 8-bit Matrix Multiplication for Transformers at Scale
>
> | Pruning Ratio | #Param | Memory     | Latency | Speedup | BoolQ | PIQA  | HellaSwag | WinoGrande | ARC-e | ARC-c | OBQA  | Average |
> |---------------|--------|------------|---------|---------|-------|-------|-----------|------------|-------|-------|-------|---------|
> | LLaMA-7B      | 6.74B  | 12884.5MiB | 69.32s  | 1x      | 73.18 | 78.35 | 72.99     | 67.01      | 67.45 | 41.38 | 42.40 | 63.25   |
> | LLM.int8()    | 6.74B  | 6777.7MiB  | 76.20s  | 0.91x   | 73.36 | 78.18 | 73.01     | 66.93      | 67.47 | 40.87 | 41.80 | 63.09   |
> | LLaMA-5.4B    | 5.47B  | 10488.4MiB | 58.55s  | 1.18x   | 76.57 | 77.37 | 66.60     | 65.82      | 70.62 | 40.70 | 38.80 | 62.36   |

---

> > ### Comment · Reviewer_VdKx · 2023-08-20
> >
> > Thanks for your detailed explanation and supplemented experiments. Will keep my score. Hope you can add these results in your final version.

---

> > > ### Author Response · Authors · 2023-08-20
> > >
> > > Dear Reviewer VdKx,
> > >
> > > Thanks for your valuable feedback. We will include these two experiments (about Quantization and Task-specific pruning) in the updated version.
> > >
> > > Best Regards,
> > > Authors of Submission 2474

---

### Official Review · Reviewer_3DoF · 2023-07-07

**Soundness:** 2 fair
**Presentation:** 3 good
**Contribution:** 3 good
**Rating:** 4
**Confidence:** 4

**Summary:**

This paper present a method for LLM pruning, where some parameters are discarded by sacrificing the downstream accuracy.
The proposed method, LLM-Pruner, adopts structural pruning that selectively removes non-critical coupled structures based on gradient information.
By investigating different pruning settings with different LLMs, this paper finds the proposed method can help reduce the computations while retaining the zero-shot capabilities.

**Strengths:**

- The proposed method is task-agnostic, not relying on domain/downstream data.
- Experimented with three different LLMs, with consistent observations.

**Weaknesses:**

- Missing the comparison between a pruned larger model and a smaller unpruned model. For example, when pruning 50% parameters for 7B model, how the results compared with a 3B model?
- Examining the proposed method with different models is good. However, instead of experimenting with 3 LLMs in similar sizes (7B, 7B, and 6B parameters), pruning a larger one would be more important. For example, a 13B llama/Vicuna model.
- Experiments only cover the proposed method, weakly connected to other model compression works.

**Questions:**

Is the proposed method a general model compression method or designed for LLM? After reading this submission, my feeling is this is general method while the method's name is LLM-pruner.

---

> ### Author Rebuttal · Authors · 2023-08-09
>
> We truly appreciate the reviewer for the constructive comments.
>
> ---
>
> > W1: Missing the comparison with the smaller models. For example, when pruning 50\% parameters for 7B model, how the results compared with a 3B model?
>
> Thanks for the constructive suggestion. As advised, we show the experimental results comparing the popular StableLM-3B and ours below. Specifically, for the sake of fairness, we selected the `stabilityai/stablelm-tuned-alpha-3b`, since it has also been tuned on Alpaca.  The performances of the two models are shown in the table below.  All results are obtained through the `lm-evaluation-harness`. LLaMA-3B obtained by LLM-Pruner demonstrates superior performance compared to StableLM-3B, incurring only a minor expense (with 7.3M vs. 800B tokens for training) to attain this compact LLM, as opposed to undertaking a complete retraining process. We would include these results in the revised version.
>
> | Pruning Ratio | #Param | Latency | #Tokens for Training | BoolQ | PIQA  | HellaSwag | WinoGrande | ARC-e | ARC-c | OBQA  | Average |
> |----|--------|------|------|-------|-------|-----|------|-------|-------|-------|---------|
> | StableLM-3B   | 3.6B   | 31.69s  | 800 Billion         | 48.78 | 69.48 | 44.52     | 54.62      | 50.93 | 25.17 | 27.40 | 45.84   |
> | LLaMA-3B  by LLM-Pruner    | 3.6B   | 37.96s  | 7.3 Million            | 61.41 | 70.08 | 51.01     | 55.01      | 46.80 | 30.38 | 37.40 | 50.30   |
>
> ---
>
> > W2: Instead of experimenting with 3 LLMs in similar size (7B, 7B, and 6B parameters), pruning a larger one would be more important. For example, a 13B llama/Vicuna model.
>
> Thanks. We show the results using LLM-Pruner to prune 20% parameters of LLaMA-13B here:
>
> | Pruning Ratio        | Method    | WikiText2 | PTB    | BoolQ | PIQA  | HellaSwag | WinoGrande | ARC-e | ARC-c | OBQA  | Average |
> |----------|---------|-----------|--------|-------|-------|-----------|------------|-------|-------|-------|---------|
> | Ratio = 0%            | LLaMA-13B | 11.58     | 20.24  | 68.47 | 78.89 | 76.24     | 70.09      | 74.58 | 44.54 | 42.00 | 64.97   |
> | Ratio = 20% w/o tune  | L2        | 61.15     | 91.43  | 61.50 | 67.57 | 52.90     | 57.54      | 50.13 | 31.14 | 36.80 | 51.08   |
> |                      | Random    | 19.24     | 31.84  | 63.33 | 73.18 | 63.54     | 60.85      | 64.44 | 36.26 | 38.00 | 57.09   |
> |                      | Channel   | 49.03     | 106.48 | 62.39 | 66.87 | 49.17     | 58.96      | 49.62 | 31.83 | 33.20 | 50.29   |
> |                      | Block     | 16.01     | 29.28  | 67.68 | 77.15 | 73.41     | 65.11      | 68.35 | 38.40 | 42.40 | 61.79   |
> | Ratio = 20% w/ tune  | L2        | 20.97     | 38.05  | 73.24 | 76.77 | 71.86     | 64.64      | 67.59 | 39.93 | 40.80 | 62.12   |
> |                      | Random    | 16.84     | 31.98  | 64.19 | 76.06 | 68.89     | 63.30      | 66.88 | 38.31 | 40.80 | 59.78   |
> |                      | Channel   | 17.58     | 29.76  | 69.20 | 76.55 | 68.89     | 66.38      | 62.08 | 38.99 | 39.60 | 60.24   |
> |                      | Block     | 15.18     | 28.08  | 70.31 | 77.91 | 75.16     | 67.88      | 71.09 | 42.41 | 43.40 | 64.02   |
>
> Most of the observations align with the experimental findings on LLaMA-7B.
> Notably, when contrasting the experimental outcomes of LLaMA-7B and LLaMA-13B, it becomes evident that the parameter redundancy in LLaMA-13B is higher than that in LLaMA-7B. Following the removal of 20% of the parameters, LLaMA-7B exhibited a reduction of 5.86%. By contrast, LLaMA-13B experienced a decrease of merely 3.18%. Furthermore, after undergoing fine-tuning, LLaMA-13B demonstrated a minor decline of only 0.95% (from 64.97 to 64.02), which significantly outperforms LLaMA-7B's decline of 3.43% (from 63.25 to 59.82).
>
> ---
>
> > W3: Experiments only cover the proposed method with a weak connection to other model compression work.
>
> Thanks. LLM-Pruner focuses on developing an efficient model compression algorithm for LLMs, with reduced dependence on data. The reason we don't extend previous approaches to LLM as a baseline is their high demand for large-scale datasets for compression. Previous methods, like those in knowledge distillation, require the training dataset to recover the model performance. For instance, LLaMA's training set has 1T tokens, which is often impractical for compressing due to high costs.
>
> Below, we conducted an experiment utilizing DistilBERT as the baseline to demonstrate the performance when applying the same dataset to compress LLaMA-7B using both LLM-Pruner and DistilBERT. We employed the same dataset, Alpaca, as the dataset for the recovery stage. A comparison between a 50% pruned model through LLM-Pruner and DistilBERT highlights LLM-Pruner's superior performance. This advantage arises from LLM-Pruner's enhanced capability to preserve critical knowledge from the base model, resulting in reduced data dependency.
>
> | Pruning Ratio | #Param | BoolQ | PIQA  | HellaSwag | WinoGrande | ARC-e | ARC-c | OBQA  | Average |
> |---|---|---|---|------|------|----|----|---|-----|
> | DistilBert    | 3.50B  | 54.77 | 63.28 | 40.44     | 52.49      | 43.31 | 27.22 | 31.00 | 44.64   |
> | LLM-Pruner    | 3.35B  | 61.56 | 68.72 | 46.62     | 52.64      | 47.94 | 29.27 | 35.40 | 48.88   |
>
> ---
>
> > Q1: Is the proposed method a general model compression method or designed for LLM? After reading this submission, my feeling is this is general method while the method's name is LLM-pruner
>
> We thank the reviewer for the comment. The proposed strategy is indeed general, yet tailored for LLM with dedicated designs. For instance, we skip some initial and final layers, adapt different pruning strategies during the pruning process and employ LoRA for fine-tuning. These design choices are incorporated to ensure the model's performance is retained as much as possible when pruning and facilitate model recovery with limited data. These designs are unnecessary when data is abundant and model scale is less of a concern.

---

> > ### Author Response · Authors · 2023-08-21
> >
> > Dear Reviewer 3DoF:
> >
> > We truly want to hear from you as the end of the discussion is approaching. Your constructive comments on our work are crucial, and we genuinely hope to get feedback from you before the discussion period ends.
> >
> > Regarding the weaknesses you mentioned, we include the corresponding experiments as follows:
> > 1. Missing the comparison between a pruned larger model and a smaller unpruned model -> We add an experiment comparing StableLM-3B with the pruned model produced by our method.
> > 2. Pruning a larger one would be more important -> We provide the pruning results on LLaMA-13B.
> > 3. Weakly connected to other model compression works -> We add an experiment to compare our method with DistilBERT.
> >
> > We truly appreciate your time and feedback on our rebuttal.
> >
> > Best Regards,
> > Authors of submission 2474

---

> > ### Comment · Reviewer_3DoF · 2023-08-21
> >
> > Thanks for the update!
> >
> > My remaining questions are:
> > - Is StableLM-7B and LLaMA-7B comparing apple-to-apple? If not, comparing StableLM-3B with LLaMA-3B by LLM-Pruner makes no sense to me...
> > - LLama 13B's results are worse than the official results, for example, LLama 13B reports boolQ with 78.1 while here is 68.4. I understand the prompt may be different but 10 percent is a huge difference.

---

> > > ### Author Response · Authors · 2023-08-21
> > >
> > > Dear Reviewer 3DoF:
> > >
> > > > Q1: Is StableLM-7B and LLaMA-7B comparing apple-to-apple? If not, comparing StableLM-3B with LLaMA-3B by LLM-Pruner makes no sense to me...
> > >
> > > Thanks for the valuable comment. As pointed out by the reviewer, StableLM-7B is not a perfect apple-to-apple comparison with LLaMA-7B. We select StableLM since they share a similar structure and have training corpora of similar size with LLaMA. Despite the inevitable gap between the two LLMs, we want to use this experiment to support the claim that our approach crafts lightweight LLMs with low resources, and even can sometimes achieve better performance than LLMs from scratch training. For example, the pruned LLaMA-3B only requires 7.3 Million tokens compared to the 800 Billion of StableLM-3B, with ~100,000$\times$ difference in corpus size. However, if an official LLaMA-3B is available, we also acknowledge that the LLaMA-3B obtained by LLM-Pruner may not outperform the LLaMA-3B from scratch training, due to the huge gap in the size of training data. We agree that “Scratch Training vs. Pruning” is an important topic [1] in the field of pruning and will follow your advice to incorporate the discussion and experiments about this in the main body of the revised version.
> > >
> > > [1] Rethinking the Value of Network Pruning. Zhuang Liu, Mingjie Sun, Tinghui Zhou, Gao Huang, Trevor Darrell
> > >
> > > ---
> > >
> > > > Q2: LLama 13B's results are worse than the official results, for example, LLama 13B reports boolQ with 78.1 while here is 68.4. I understand the prompt may be different but 10 percent is a huge difference.
> > >
> > > Yes. This problem is caused by the difference in prompts, and this issue has also been discussed in previous works. For example, [2] mentioned that *"Note: We are using lm-eval-harness to evaluate our performance. Therefore, LLaMA numbers are not supposed to be compared from the original paper since we are using different methods of measurement"*. This issue is also discussed in the GitHub repo(EleutherAI/lm-evaluation-harness). We follow the official instruction and also its prompt to evaluate the LLaMA-13B. All of our experiments use the same set of prompts to ensure fairness.
> > >
> > > [2] LaMini-LM: A Diverse Herd of Distilled Models from Large-Scale Instructions. Minghao Wu, Abdul Waheed, Chiyu Zhang, Muhammad Abdul-Mageed, Alham Fikri Aji
> > >
> > > Best Regards,
> > > Authors of Submission 2474

---

> > > > ### Comment · Reviewer_3DoF · 2023-08-21
> > > >
> > > > Thanks for your response!
> > > >
> > > > For Q1:
> > > >
> > > > I would suggest to use Stability 7B and Stability 3B for this experiment. It is okay to show pruned-Stability-3B is worse than Stability-3B. This results can help understand the advantage/limitation of the proposed method and inspire future work. Also, I think this claim "pruned LLaMA-3B only requires 7.3 Million tokens compared to the 800 Billion of StableLM-3B" is not accurate, as pruned-llama-3b is 7.3M + (pretrained tokens of LLama-3B) instead of 7.3M.
> > > >
> > > > For Q2:
> > > >
> > > > I understand the Q2 is caused by different prompting and `lm-evaluation-harness` is definitely a choice. However, if the gap between LLama's paper and `lm-evaluation-harness` is huge, I would suggest to also report LLama's prompting results. Although LLama did not publish its prompts, LLama described what prompts they were using. For example, they said: for OpenBookQA, BoolQ they follow GPT-3's template; for other tasks, they follow  Gao et al. (2021) and use the likelihood normalized by the number of characters in the completion.
> > > >
> > > > The reason I concern Q2 is because different prompting can lead to huge differences and randomness. Although in this submission, the comparisons are evaluated with the same prompting method, evaluating with a same SOTA prompting method v.s. with a same un-SOTA prompt method are still different. I appreciate the current  `lm-evaluation-harness`  results and would suggest to add another table (in appendix) to show your efforts to reproduce llama's results with llama's prompts, and show your pruned results.
> > > >
> > > > Thank you again.

---

> > > > > ### Author Response · Authors · 2023-08-21
> > > > >
> > > > > > I would suggest to use Stability 7B and Stability 3B for this experiment. It is okay to show pruned-Stability-3B is worse than Stability-3B.
> > > > >
> > > > > Thanks for the suggestion. We will provide more results on StableLM. Since it takes time to conduct these experiments, like searching the hyperparameters and evaluating the models, we cannot provide it here. It would be better if we could discuss it earlier so that we can provide the results in the discussion period to fully address your concern.
> > > > >
> > > > > > Also, I think this claim "pruned LLaMA-3B only requires 7.3 Million tokens compared to the 800 Billion of StableLM-3B" is not accurate, as pruned-llama-3b is 7.3M + (pretrained tokens of LLama-3B) instead of 7.3M.
> > > > >
> > > > > We kindly remind the reviewer that this work is about model compression and thus we may focus on the compression cost. For instance, if we need a 5B model, the training cost of LLM-Pruner is only 7.3M corpus, while other feasible ways like pre-training an LLM from scratch require 800 billion tokens.
> > > > >
> > > > > > The reason I concern Q2 is because different prompting can lead to huge differences and randomness. Although in this submission, the comparisons are evaluated with the same prompting method, evaluating with a same SOTA prompting method v.s. with a same un-SOTA prompt method are still different.
> > > > >
> > > > > Thanks for your understanding that we cannot get the prompt of LLaMA since they didn't release the prompt. We will try to guess the prompt of the official LLaMA. If we find some useful results, we would report them in the appendix.
> > > > >
> > > > > And we also want to clarify that our focus is the relative performance of the pruned and the un-pruned model. To ensure fairness and make the evaluation a **standardized process**, a common practice is to use the official prompt of lm-eval-harness.

---

### Official Review · Reviewer_ahc6 · 2023-07-10

**Soundness:** 3 good
**Presentation:** 3 good
**Contribution:** 2 fair
**Rating:** 6
**Confidence:** 4

**Summary:**

This work proposes an LLM compression framework, LLM-Pruner, that employs structural pruning to compress LLMs in a task agnostic manner so that their multi-task solving is preserved. They achieve this without using the original training data. In addition, they employ adaptors (LoRA) during the tuning phase for parameter and training efficiency. They evaluate the models on zero-shot classification and text generation and demonstrate reasonable performance compared to the original model.

**Strengths:**

1. Although task agnostic knowledge distillation exists (TinyBert, DistilBERT etc.) and can be done without the original training data, it requires a significant training effort. This work can achieve upto 5 times compression with minimal training.

2. The estimation of group importance estimation is important for gaining an inference speed advantage for pruning. However, the authors did not present an analysis on the impact of their method on inference speed.

**Weaknesses:**

1. Although the authors presents the reduction in parameters, they do not mention the inference speedup with and without LLM-pruner. What would be specially interesting would be the inference speed under different types of group pruning (e.g. channel vs block)

**Questions:**

1. LoRA and other adaptors are not necessarily faster to train. They reduce the number of trainable parameters but the backprop generally runs through the entire network. Can you compare the training time with and without LoRA (i.e. full network training).

2. The Ablation Section at the beginning of line 227 explains a table but does not refer to one. I assumed it was Table 3 but if so some of the explanations were not clear.

3. Weights and parameters are some times used interchangeably in deep learning. What is meant by weights in Section 3.2.

**Limitations:**

Yes.

---

> ### Author Rebuttal · Authors · 2023-08-09
>
> We truly appreciate the reviewer for the constructive comments.
>
> >W1: Although the authors presents the reduction in parameters, they do not mention the inference speedup with and without LLM-pruner. What would be specially interesting would be the inference speed under different types of group pruning (e.g. channel vs block)
>
> We thank the reviewer for the valuable suggestion. In the table below, we show the latency of different grouping strategies on LLaMA-7B. The latencies shown in the table correspond to runtime evaluations on the test set of WikiText2 and PTB. We test the result on A5000 with 24GB memory. Notably, both strategies accelerate the inference over LLaMA-7B. Nevertheless, a performance disparity persists between these two strategies, with the block strategy outpacing the channel strategy in speed. Such discrepancy arises from distinct pruning approaches, resulting in variations in the model structure.This presents an intriguing topic for further investigation, and we will add these results to the revised version.
>
> | Group   | Ratio | #Params | Memory     | Latency(WikiText2) | Latency(PTB) |
> |---------|-------|---------|------------|---------------|--------------|
> | /      | 0%     | 6.74B   | 12884.5MiB | 69.32s        | 20.47s       |
> | channel | 20\%  | 5.47B   | 10488.4MiB | 61.50s        | 18.30s       |
> | block   | 20\%  | 5.42B   | 10375.5MiB | 58.55s        | 17.55s       |
> | channel | 50\%  | 3.37B   | 6556.3MiB  | 40.11s        | 11.96s       |
> | block   | 50\%  | 3.35B   | 6533.9MiB  | 37.54s        | 11.27s       |
>
> > Q1: LoRA and other adaptors are not necessarily faster to train. They reduce the number of trainable parameters but the backprop generally runs through the entire network. Can you compare the training time with and without LoRA (i.e. full network training).
>
> Thanks for the comment. The table below shows the contrast between training time and accuracy achieved through LoRA and full-parameter tuning obtained on A5000 GPUs with 24G memory, where LoRA indeed demonstrates faster training.
> This is mainly due to two reasons: (1) Constraints in conducting full-parameter fine-tuning on a single 24G GPU result in increasing communication overhead across multiple GPUs. (2) The greater quantity of parameter updates, along with additional updates in the optimizer during full-parameter training, leads to prolonged training durations.
>
> Apart from comparing the time, we also show the accuracy with and without LoRA.
> our results indicate LoRA's superior performance compared to full-parameter supervised fine-tuning in this case. This is attributed to the dataset's small size, making full-parameter training susceptible to overfitting, especially in the context of zero-shot evaluation of LLMs.
>
> | Pruning Ratio      | Time      | #GPUs | BoolQ | PIQA  | HellaSwag | WinoGrande | ARC-e | ARC-c | OBQA  | Average |
> |--------------------|-----------|-------|-------|-------|-----------|------------|-------|-------|-------|---------|
> | Full-Parameter SFT | 5.4 hours | 8     | 62.20 | 62.68 | 41.31     | 53.83      | 48.19 | 29.35 | 33.60 | 47.31   |
> | LoRA               | 3.5 hours | 1     | 61.56 | 68.72 | 46.62     | 52.64      | 47.94 | 29.27 | 35.40 | 48.88   |
>
> > Q2: The Ablation Section at the beginning of line 227 explains a table but does not refer to one. I assumed it was Table 3 but if so some of the explanations were not clear.
>
> We thank the reviewer for the comments and truly apologize for the missing reference for the table. The ablation experiment mentioned in Line 227 refers to Table 3, with the line annotated with "parameter" or "weight" (The names would also be modified as described in the next question.)
>
> > Q3: Weights and parameters are sometimes used interchangeably in deep learning. What is meant by weights in Section 3.2.
>
> We thank the reviewer for the comment. Indeed, the wording of `weights` and `parameters` could cause ambiguity, for which we apologize. In our revised version, we will change our terminologies to `vector-wise importance` (previously `weight`)
> and `element-wise importance` (previously `parameter`). The two represent different importance-assessment granularities: `Vector-wise importance` involves evaluating the entirety of weight parameters within the sample module as a whole through Taylor expansion, while `element-wise importance` pertains to assessing each individual weight in the same module distinctly.

---

> > ### Comment · Reviewer_ahc6 · 2023-08-20
> >
> > Thank you for sharing these additional results. I have updated my score accordingly.

---

> > > ### Author Response · Authors · 2023-08-20
> > >
> > > Dear Reviewer ahc6,
> > >
> > > Thanks again for your valuable comments. We will follow the advice and add the above results, especially the latency analysis, into the updated version.
> > >
> > > Best Regards,
> > > Authors of Submission 2474

---

### Author Response · Authors · 2023-08-20
**Looking forward to further discussion**

Dear Reviewers,

We sincerely appreciate your valuable and insightful comments. Noticing that there are only around 35 hours left until the end of the discussion period, we kindly ask for a few minutes of your time to check our response. We look forward to the opportunity to discuss this with you. Hearing your further opinion about this would be essential to us to improve our work.

Here we briefly summarise our response with respect to the concerns you mentioned in the weaknesses or questions:

* **To Reviewer ahc6**: We include a latency analysis of different pruning strategies and a comparison between LoRA and full-parameter tuning.
* **To Reviewer 3DoF**: Our response provides (1) an evaluation of models pruned by LLM-Pruner versus the StableLM-3B, (2) the results of LLM-Pruner on LLaMA-13B, and (3) the addition of another baseline in KD (DistilBERT), as per the weaknesses you listed.
* **To Reviewer VdKx**: We conduct experiments about task-specific pruning and quantization, and provide an analysis of these results.
* **To Reviewer mPK3**: We show that the pruned model is not a sparse model and include a latency comparison between the pruned model and the original model. Besides, we compare our pruned model with StableLM-3B.

We deeply appreciate your thoughtful comments and the time you have dedicated to reviewing our paper.

Best regards,
Authors of submission 2474

---

### Decision · Program_Chairs · 2023-09-21

**Decision:**

Accept (poster)

**Comment:**

The paper addresses a very important problem: pruning an LLM to a smaller, more efficient, and yet well performing model. The method is certainly useful, and the experiments are generally comprehensive. However, there are some important baselines that are not included in the submitted version of the paper.

The authors did an amazing job in their rebuttals, conducting many new experiments to address the reviewers concerns, and do that satisfactorily in my opinion. But these are the results not currently in the paper. If they are added, the paper will present a comprehensive evaluation. I, therefore, recommend conditional acceptance of this paper provided that the new experiments from the rebuttal are adequately incorporated.